# A cleaner snow future mitigates Northern Hemisphere snowpack loss from warming

Dalei Hao [1] ✉, Gautam Bisht[1], Hailong Wang [1], Donghui Xu [1], Huilin Huang[1], Yun Qian [1] & L. Ruby Leung [1] ✉

Light-absorbing particles (LAP) deposited on seasonal snowpack can result in snow darkening, earlier snowmelt, and regional climate change. However, their future evolution and contributions to snowpack change relative to global warming remain unclear. Here, using Earth System Model simulations, we project significantly reduced black carbon deposition by 2081-2100, which reduces the December-May average LAP-induced radiative forcing in snow over the Northern Hemisphere from 1.3 Wm$^{-2}$ during 1995-2014 to 0.65 (SSP126) and 0.49 (SSP585) Wm$^{-2}$. We quantify separately the contributions of climate change and LAP evolution on future snowpack and demonstrate that projected LAP changes in snow over the Tibetan Plateau will alleviate future snowpack loss due to climate change by 52.1 ± 8.0% and 8.0 ± 1.1% at the end of the century for the two scenarios, mainly due to reduced black carbon contamination. Our findings highlight a cleaner snow future and its benefits for future water supply from snowmelt especially under the sustainable development pathway of SSP126.

Seasonal snow plays a critical role in Earth's energy budget and water cycle[1], with snowmelt over mountainous regions providing an important source of freshwater for two billion people globally[2,3]. However, anthropogenic climate change is projected to reduce global snowfall and cause earlier snowmelt[4]. The high-mountain Asia (HMA), known as the water tower of Asia, has been experiencing an overall decrease in snow water equivalent (SWE) in the last 30 years[5,6]. The Western United States (WUS) is also expecting a low-to-no snow future, with an SWE decline of ~25% by 2050[7]. Projections show a continuous decline in SWE for nearly all global high-mountain regions throughout the 21st century under both the low and high Representative Concentration Pathway (i.e., RCP2.6 and RCP8.5) greenhouse gas emission scenarios[8]. Future snowpack changes under global warming will significantly alter downstream runoff as well as the amount and timing[9] of freshwater supply from snowmelt for both humans and ecosystems.

Light-absorbing particles (LAP), such as black carbon (BC) and dust, can darken snow surface, reduce snow albedo, and accelerate snowmelt processes[10,11]. BC originates from the incomplete burning of fossil fuels, biomass, or biofuels[12], while dust can be produced from the natural wind erosion of soil or anthropogenic industrial emissions[13]. Snow darkening due to BC has been demonstrated to significantly contribute to historical and present-day snowmelt[14] and climate change[15]. Dust was found to have larger darkening impacts than BC over high-altitude regions of HMA[10]. However, the future evolution of LAP deposition and the associated snow-darkening impacts have not been quantified hitherto.

A clean or dirty snow future could have a different impact on snowmelt and snow water resources. Future BC emissions are projected to decrease in both low and high-emission scenarios, but the spatial and temporal patterns are expected to vary among different scenarios[16]. Under RCP8.5, most regions in the Northern Hemisphere (NH) are projected to show a significant decrease in BC deposition by 2050[17]. The intensification of human land use and drought induced by climate change will affect future dust emissions and depositions[18]. The projected 25–40% loss of biological soil crusts (i.e. biocrusts) will lead to around a 5–15% increase in global dust emission and deposition by 2070[19]. How the future change of LAP emission and deposition over snow-covered regions will impact snowpack remains uncertain.

[1]Atmospheric Sciences and Global Change Division, Pacific Northwest National Laboratory, Richland, WA, USA. ✉e-mail: dalei.hao@pnnl.gov; ruby.leung@pnnl.gov

Besides, current climate models project a significant decrease in future global snowpacks due to global warming[4,20]. However, such projections can be biased since most climate models mainly account for changes in climate (i.e., precipitation and temperature) but neglect or oversimplify the effects of snow darkening due to BC and dust deposition[8]. Understanding the relative contribution of climate change and LAP to future snowpack change with the considerations of model uncertainties is critical for robustly constraining projections of downstream freshwater availability from snowmelt.

Here we first use model outputs that are archived in the Coupled Model Intercomparison Project Phase 6 (CMIP6) to analyze the evolution of BC and dust deposition from 2015 to 2100 over the NH. We explore two contrasting Shared Socioeconomic Pathway and Representative Concentration Pathway (SSP-RCP) scenarios (i.e., SSP126 and SSP585). We then examine the future spatio-temporal characteristics of LAP mass, snow albedo reduction, and surface radiative forcing (RF, a measure of the net change in surface radiative fluxes due to the change of a forcing agent) induced by BC and dust, using the state-of-the-art Energy Exascale Earth System Model (E3SM) Land Model (ELM) driven by meteorological forcing and LAP deposition data simulated by a CMIP6 model (see the "Methods" section). We also analyze the individual impacts of BC and dust evolution and quantify separately the contributions of climate change and the evolution of LAP (i.e., BC and dust) to future snowpack change to better understand their implications for future snow water resources. We further quantify the impacts of model structural uncertainties on the projection of future snowpack changes in the NH. Our analyses mainly focus on December–May because the impacts of LAP on snow are generally larger in mid-latitude mountains than high-latitude regions[11] and these regions have no or low snow cover from June to November.

## Results

### Future evolution of BC and dust deposition

Compared to the historical period (1995–2014), BC depositions over the entire snow-covered regions of the NH are projected to significantly decrease by 2081–2100 under both SSP126 and SSP585 (Fig. 1a–g). The historical BC deposition rates are higher in relatively lower latitude regions and show the largest values over the Southern border of the Tibetan Plateau (TP) and East Asia (Fig. 1a). Such high BC deposition rates consist of both wet and dry depositions (Fig. S1), resulting from abundant emissions sources ranging from the significant fossil fuel combustion and traditional biomass usage in the Asian regions[16,21]. The projected future BC deposition rates exhibit a similar spatial pattern as historical rates, but with significantly smaller magnitudes (Fig. 1b, e). Compared to the historical period, SSP126 shows larger BC deposition rate decreases than SSP585, especially in western Asia and eastern North America. There are significant decreasing trends in BC deposition from 2015 to 2100 over nearly all the snow-covered regions of the NH (Figs. 1c, f, S2a, b and S3a, b). The average BC deposition rates show a significant decreasing trend ($p < 0.05$ based on the Mann–Kendall (MK) test) from 2015 to 2100, with higher decreasing rates from 2015 to 2040 (Fig. 1d). Compared to SSP126, SSP585 also shows a significant decreasing trend ($p < 0.05$ from the MK test) but at a relatively more constant rate throughout the century (Fig. 1g). By the end of the century, both SSP126 and SSP585 show a low level of BC deposition (Figs. 1b–g and S4), especially over the regions around the TP (e.g., China and India), due to the significant decline of BC emission (Fig. S5) from traditional biomass usage and transport-related activity with the increased economic development and population stabilization and emissions controls under both scenarios[16,22]. The relatively small standard deviations of the seven CMIP6 models that provide the deposition data for both BC and dust indicate high model agreement in the decreasing trends (shaded regions in Fig. 1d, g). The significant decrease in projected BC deposition can be attributed to

socioeconomic development and technological progress that have reduced BC emissions worldwide[16].

Compared to BC, overall future dust deposition is projected to be larger under both future scenarios during 2081–2100 than the historical period, especially over the TP (Fig. 1h–n). The spatial patterns of dust deposition rates are similar under the historical and future scenarios. The northern and western TP regions show large values because of their proximity to drylands, the main source of dust emission[23]. Most regions show no significant trends in dust deposition under SSP126, while many Asian regions show a significant increasing trend (MK test: $p < 0.05$) under SSP585 (Figs. S2c, d and S3c, d), due to climate change and anthropogenic land use[24–26]. The average dust deposition rates over snow-covered NH regions show large interannual variability and have a small, and insignificant increasing trend (MK test: $p > 0.05$) from 2015 to 2100 under both scenarios (Fig. 1k, n). The large inter-model differences (shaded regions in Fig. 1k, n) can be attributed to different parameterizations of near-surface winds, soil erodibility, and/or vegetation evolution (prescribed vs. dynamic vegetation) as well as diverse treatments of the size of emitted dust particles in ESMs[23,27]. Despite the different and potentially opposite trends of future BC and dust depositions, the significant reduction of BC deposition is expected to have a larger effect on snow than dust changes.

### Future surface radiative forcings from BC and dust in snow

The ELM-simulated BC and dust concentrations in the entire snow column are well correlated with the field measurements across the NH with the correlation coefficients of 0.66 and 0.74, respectively (see the "Methods" section; Fig. S6). Next, we analyze the RF of BC, dust, and LAP (i.e., the sum of BC and dust) in snow simulated by ELM (see the "Methods" section). Northern Asia and TP have large RF of BC, dust, and LAP during the historical period (Fig. 2a, d, g). Consistent with ref. 10, historically, dust can play a greater role than BC in reducing snow albedo over high-altitude regions of HMA (Fig. S7). Due to reduced BC deposition, the projected BC RF over the whole NH decreases significantly by 2081–2100 (Fig. 2a–c) compared to the historical period (Fig. 1). However, the change of dust RF is sensitive to the emission scenarios, with an increase in SSP126 but a decrease in SSP585 (Fig. 2d–f). Due to the dominating change in BC RF, the LAP-induced RF significantly decreases over the NH under both scenarios (Fig. 2g–i; Table S1). Although BC has a larger RF than dust in the historical period, the significant reduction of BC deposition leads to a larger RF of dust than BC in the future (Table S1). These RF changes are consistent with the spatio-temporal distribution of LAP mass in the snow column (Fig. S8) and LAP-induced snow albedo reduction (Fig. S9).

TP, a hotspot of climate change, is projected to experience a continuing decrease in LAP-induced RF from 2015 to 2100 (Figs. 3 and S10; Table S1). In the Control ELM simulations (see the "Methods" section), the historical (1995–2014) average LAP-induced RF over the TP is 5.1 W m$^{-2}$, while the future (2081–2100) average RF is 2.4 W m$^{-2}$ under SSP126 (Fig. 3a). This is likely due to the reduced BC deposition (Figs. 1 and S4) and snowpack under climate change. Compared to the historical period, assuming no change in future LAP deposition (see the "Methods" section) would result in a slightly decreasing average RF over the TP of 4.1 W m$^{-2}$ during 2081–2100 under SSP126 due to the faster snowmelt and reduced snowpack in warmer temperatures. For SSP585, the future average LAP-induced RF over the TP with and without future change of LAP deposition are 1.5 and 2.3 W m$^{-2}$, respectively (Fig. 3b). Notably, the future BC-induced RFs between SSP126 and SSP585 are comparable over the TP by the end of the century due to the significant decline of BC emission and deposition (Figs. S4 and S5), suggesting that the climate warming and dust deposition explain the difference of LAP-induced RF changes. Although ELM model configurations can affect the magnitude of

simulated LAP-induced RF, their impacts on the relative difference caused by future LAP changes are small (Fig. 3a, b). Future LAP changes can account for $60.5 \pm 1.9\%$ and $21.2 \pm 1.1\%$ of the decrease of future LAP-induced RF relative to the historical period for SSP126 and SSP585, respectively. Temporally, the average BC-induced RF in the ELM simulations shows a significant decreasing trend ($p < 0.05$ from the MK test) for both SSP126 and SSP585 scenarios. The trend is reduced under the assumption that future BC remains at the historical level (Fig. S10a). The dust-induced RF in SSP126 shows a slightly increasing trend ($p < 0.05$ from the MK test) that switches to a slightly decreasing trend ($p < 0.05$ from the MK test) if future dust deposition remains at the historical value (Fig. S10b). However, dust-induced RF shows a slight decreasing trend ($p < 0.05$ from the MK test) in SSP585 that becomes stronger if future dust deposition remains unchanged (Fig. S10b). For both future scenarios, the LAP-induced RF has a

significant decreasing trend that is larger than either individual trend from BC or dust (Fig. S10c). The significant reduction of future LAP-induced RF is expected to contribute to a slowdown of future snowpack loss.

## Contributions of LAP changes to future snowpack change

We further quantify contributions of LAP changes to future snowpack changes using ELM (Fig. 4). We use April SWE in the analysis, considering that seasonal snowpack generally peaks around April in many mid-latitude snow-dominated regions[28,29], which could provide useful insight into the expected spring runoff and available water resource. Similar results are obtained for the average SWE from December to May (Figs. S11 and S12). Overall, the future SWE in April over the NH is projected to decrease compared to the historical period, especially under SSP585 (Figs. 4a–c and S13). WUS will have a significant decline in SWE

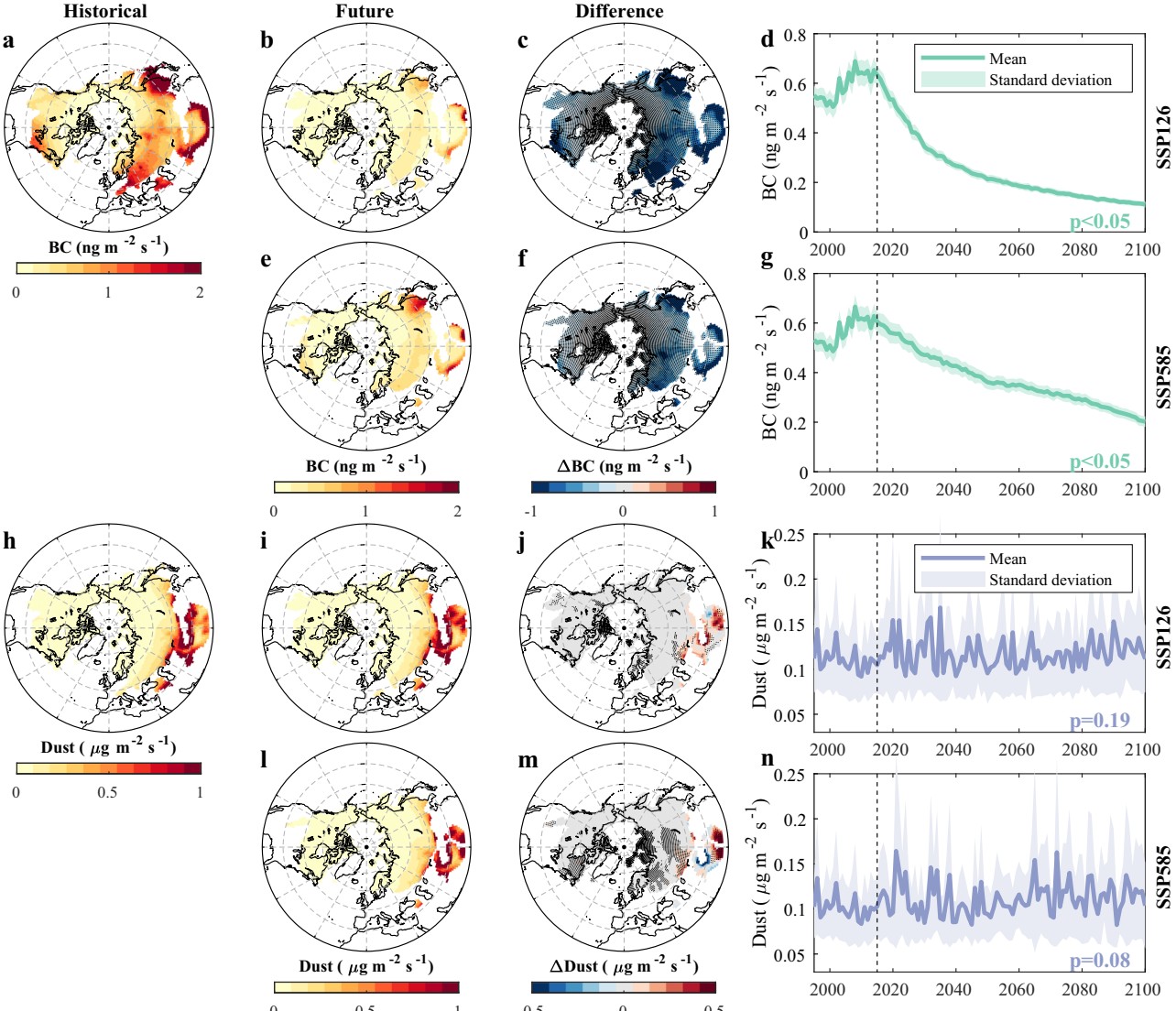

**Fig. 1 | Historical and future deposition rate of black carbon (BC) and dust.**
**a**, **h** Historical (1995–2014) and **b**, **e**, **l** future (2081–2100) spatial patterns of aerosol deposition rates and **c**, **f**, **j**, **m** their differences (calculated as Future - Historical) for BC and dust under SSP126 and SSP585. **d**, **g**, **k**, **n** Time series of the average deposition rate of BC and dust for snow-covered regions over the Northern Hemisphere (NH) where the average snow water equivalent (SWE) from December to May exceeds 5 mm. Historical and future deposition rates are calculated based on the ensemble mean of seven CMIP6 model outputs from December to May. In **a**–**c**, **e**, **f**, **h**–**j**, **i**–**m** grids with an average SWE from December to May <5 mm are

masked. In **c**, **f**, **j**, **m** the black dots represent regions with statistically significant trends ($p < 0.05$) using the Mann–Kendall (MK) test. In **d**, **g**, **k**, **n** the line and background shading represent the mean and standard deviation of deposition rates, respectively, based on the seven CMIP6 models. The $p$ values from the MK test of statistical significance of the temporal trends from 2015 to 2100 are shown inside each panel; and the vertical dashed line indicates the year 2015 when SSP scenarios start. We use ng m⁻² s⁻¹ and μg m⁻² s⁻¹ as BC and dust deposition rate units, respectively.

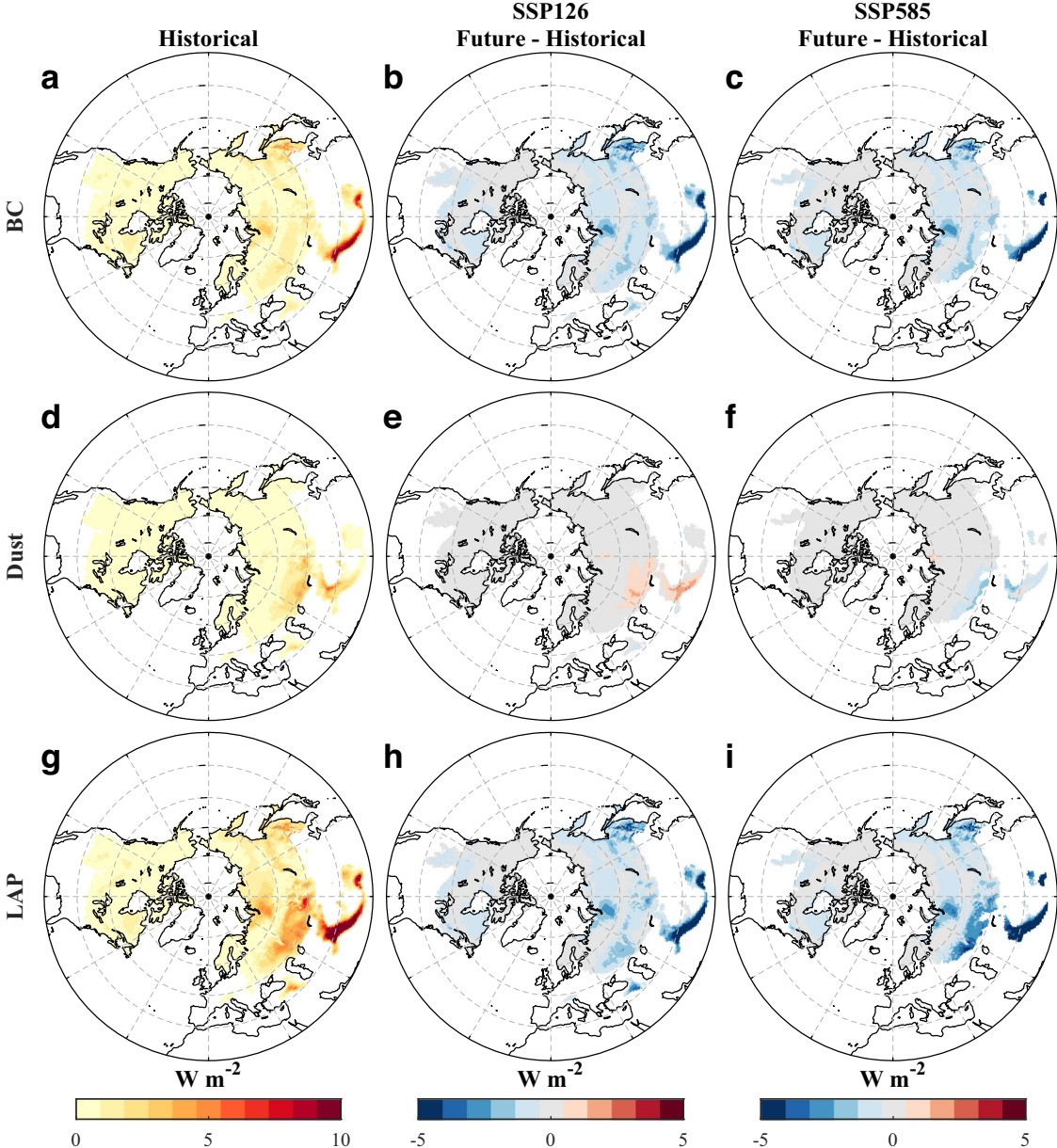

**Fig. 2 | Spatial patterns of historical and future surface radiative forcings (RF) of black carbon (BC), dust, and light-absorbing particles (LAP) (the sum of BC and dust) in snow-covered regions over the Northern Hemisphere (NH). a, d, g** Historical (1995–2014) spatial patterns of RF. **b, c, e, f, h, i** The difference between future (2081–2100) and historical RF under SSP126 and SSP585. Historical and future RF is calculated based on the Energy Exascale Earth System Model (E3SM) Land Model (ELM) outputs from December to May. In each panel, grids where the average snow water equivalent (SWE) from December to May is smaller than 5 mm are masked.

under both low and high-emission scenarios, identical to the findings of a low-to-no snow future noted in previous studies[7,30]. However, enhanced precipitation and sub-freezing temperatures will lead to an increase in SWE over the North American Arctic and high-latitude regions of Asia[31]. Consistent with previous findings[32], TP is projected to have an overall decrease in seasonal snowpack throughout the 21st century. Due to its reliance on wintertime snowfall and initial cooler summertime temperatures and a projected increase of snowfall (Fig. S14), SWE in Karakoram, located in the northwestern TP, increases (SSP126) or stays relatively stable (SSP585) in the future[33]. The projected SWE differences in Karakoram between the two scenarios are possible because SSP585 projects significantly warmer air temperature and thus faster snowmelt than SSP126 (Fig. S14h–i). We define the contribution of LAP change to snowpack change ($\Delta SWE_{LAP}$) as the difference between the projected SWE with and without future LAP change (see the

"Methods" section). $\Delta SWE_{LAP}$ is positive over the NH, especially under SSP126, and TP has the largest $\Delta SWE_{LAP}$ (Fig. 4d, e). This highlights that future LAP change will contribute to a slowdown of warming-driven snowpack loss. The relative $\Delta SWE_{LAP}$ (calculated as the ratio of $\Delta SWE_{LAP}$ to projected SWE without LAP change) over the TP significantly increases (MK test: $p < 0.05$) from 0% to over 10% (SSP126) and 20% (SSP585) during 2015–2100 (Fig. 4f, g). The future reduction of BC concentration dominates the change of relative $\Delta SWE_{LAP}$ (Fig. S10f, g), accounting for most of the slowdown of SWE reduction.

We further analyze the relative contribution of climate change (i.e., temperature and precipitation) and LAP change as well as the impacts of ELM model configurations (Figs. 3c, d and S15) on SWE. Compared to the historical period, SSP585 shows a stronger decrease in SWE than SSP126 regardless of model configuration (Fig. 3c, d). Although topography, snow grain shape, mixing states of LAP-snow,

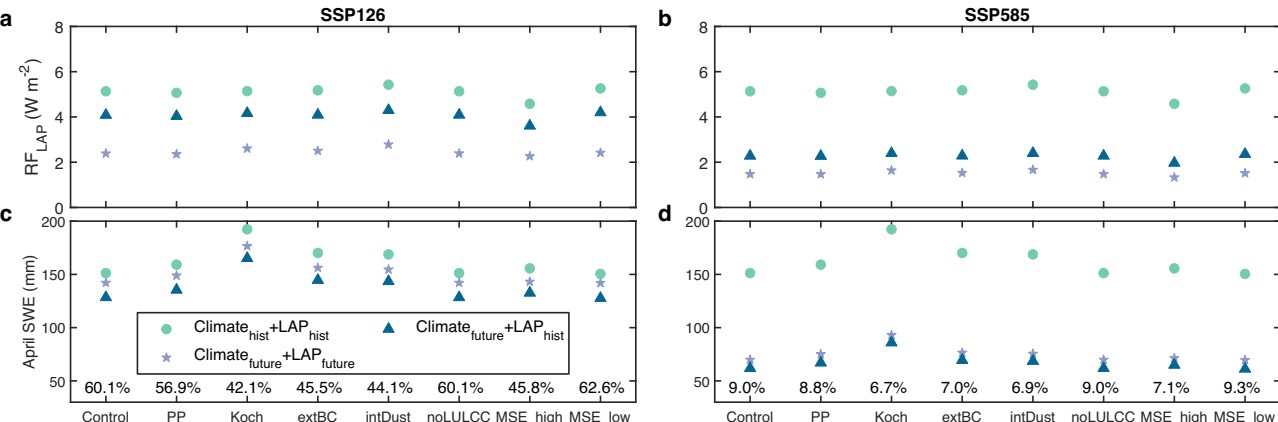

**Fig. 3 | Historical and future average surface radiative forcing (RF) induced by light-absorbing particles (LAP) and April snow water equivalent (SWE) over the snow-covered Tibetan Plateau (TP) regions (where the average SWE exceeds 5 mm in the historical period of 1995–2014) under different model configurations. a**, **b** The average RF from December to May under SSP126 and SSP585. **c**, **d** April SWE under SSP126 and SSP585. For each panel, Climate$_{hist}$ + LAP$_{hist}$ represents the historical (1995–2014) simulations with historical LAP depositions, while Climate$_{future}$ + LAP$_{future}$ and Climate$_{future}$ + LAP$_{hist}$ represent future (2081–2100) simulations with and without a future change of LAP depositions, respectively. The Climate$_{future}$ + LAP$_{hist}$ simulations used the historical average LAP depositions from 1995 to 2014. The horizontal axis labels represent different model configurations (see the "Methods" section), where Control has the Energy Exascale Earth System Model (E3SM) Land Model (ELM) default settings and the others

represent major adjustments made from the Control case. Specifically, PP assumes that the terrain is flat and neglects topographic effects on solar radiation; Koch assumes a non-spherical snow grain shape (Koch snowflake); extBC assumes external mixing between hydrophilic BC and snow grains; intDust assumes internal mixing between dust and snow grains; noLULCC has no land use and land cover change; MSE_high assumes high melt-water scavenging efficiency (MSE = 2, much higher than the default value of 0.2) of hydrophilic BC; and MSE_low assumes a low MSE (0.02) of hydrophilic BC. In **c**, **d** the contribution ($\delta_{LAP}$) of future LAP change that mitigates snowpack loss under each ELM configuration is noted as a percentage and is calculated as the ratio of the SWE difference ($\Delta SWE_{LAP}$) between Climate$_{future}$ + LAP$_{future}$ and Climate$_{future}$ + LAP$_{hist}$ to the SWE difference ($\Delta SWE_{Climate}$) between Climate$_{hist}$ + LAP$_{hist}$ and Climate$_{future}$ + LAP$_{future}$. The geographical coverage of the TP is shown in Fig. 4.

land use, and land cover changes, and melt-water scavenging efficiency can affect the magnitude of SWE, they have small impacts on the relative change of SWE (Fig. 3c, d). For example, the larger snow albedo of non-spherical snow grain shape can lead to larger SWE[34], but snow grain shape has little impact on the relative change of SWE. We further estimate climate change impacts ($\Delta SWE_{Climate}$) on SWE as the difference between the historical SWE and future SWE without LAP change. We define the contribution ($\delta_{LAP}$) of future LAP change that mitigates snowpack loss as the ratio of $\Delta SWE_{LAP}$ to $\Delta SWE_{Climate}$ (see the "Methods" section). The spatial distribution of $\delta_{LAP}$ shows that TP is the most sensitive to LAP change (Fig. S15). Under different model configurations, $\delta_{LAP}$ of TP is 52.1 ± 8.0% (SSP126) and 8.0 ± 1.1% (SSP585) (Fig. 3c, d). These results suggest that under SSP585, climate change dominates the snowpack loss and only 8% of this loss can be mitigated by LAP change, while under SSP126, LAP change can offset around 52.1% of the impacts of climate change on the TP snowpack.

## Discussion

Research has shown that LAP contributes to the accelerated melting of snow historically[10,14]. We show future decreasing trends of LAP deposition, mass, and RF under both low (SSP126) and high (SSP585) emission scenarios using simulation results from CMIP6 and ELM. Our study highlights a cleaner snow future due to the significant reduction of BC deposition on the snow surface. The projected cleaner snow will help alleviate future snowpack loss induced by warmer climates, especially over the TP. This is consistent with a recent study[35] showing that BC deposition decrease since the 1980s has moderated the influence of climate change in the decline of snow cover over the French Alps and the Pyrenees. Consistent with ref. 36, the ELM simulations show that the historical BC deposition can contribute to the SWE reduction of more than 25 mm over the TP (Fig. S16). The compensating effects of decreasing BC will also contribute to the shift in snowmelt timing and substantially influence meltwater runoff[35]. Considering the important role of the TP in Asia's freshwater supply[32], the increased SWE due to cleaner snow will be beneficial for the municipal,

hydropower[37], and agricultural[38] sectors in Asian regions as well as vegetation growth and animal survival. Cleaner snow can mitigate over half of the snowpack loss caused by climate change under the low-emission scenario (i.e., SSP126). These results stress the importance of reducing combustion aerosol emissions by developing clean, renewable energy and negative-emission technologies, in addition to mitigating climate change. However, climate factors dominate snowpack loss under the worst emission scenario (i.e., SSP585). The change in the relative role of reduced LAP highlights the necessity of constraining global warming levels to mitigate snowpack loss. Compared to the high fossil-fueled development pathway (i.e., SSP585), a sustainable, green development pathway (i.e., SSP126) will better benefit future water supply from snowmelt.

Beyond the significant decrease of RF of BC over the TP, the RF of dust will increase under SSP126. This is potentially due to changing meteorological conditions (e.g., precipitation, humidity, wind) and increasing aridity as well as continued land use and land cover change[24]. The dust RF is projected to have a small decrease under SSP585 (Fig. 2). Consequently, dust is projected to account for a larger portion of LAP-induced RF than BC in the future under both SSP126 and SSP585, while BC accounts for more in the historical period (Figs. 2 and S10; Table S1). The increasing contributions of dust to snowmelt will be more significant over the high-altitude HMA, as previous work demonstrated that dust dominates high-altitude snow darkening and melting over HMA[10]. These highlight the importance of mitigating soil disturbance and stabilizing soil surface in dust source regions to reduce both natural and anthropogenic dust emissions[39].

However, there remain uncertainties in simulating the LAP darkening effects on snow. First, accurately characterizing the emission, transport, chemistry, and deposition of LAP under changing climate is challenging. The emission, cycling, and persistence of BC are still under-represented in ESMs[40]. Although reduced fossil fuel burning in developing countries can reduce future BC emissions, increased wildfire intensity and frequency due to climate change and land-use change may potentially increase the BC emissions[41]. However, ESMs

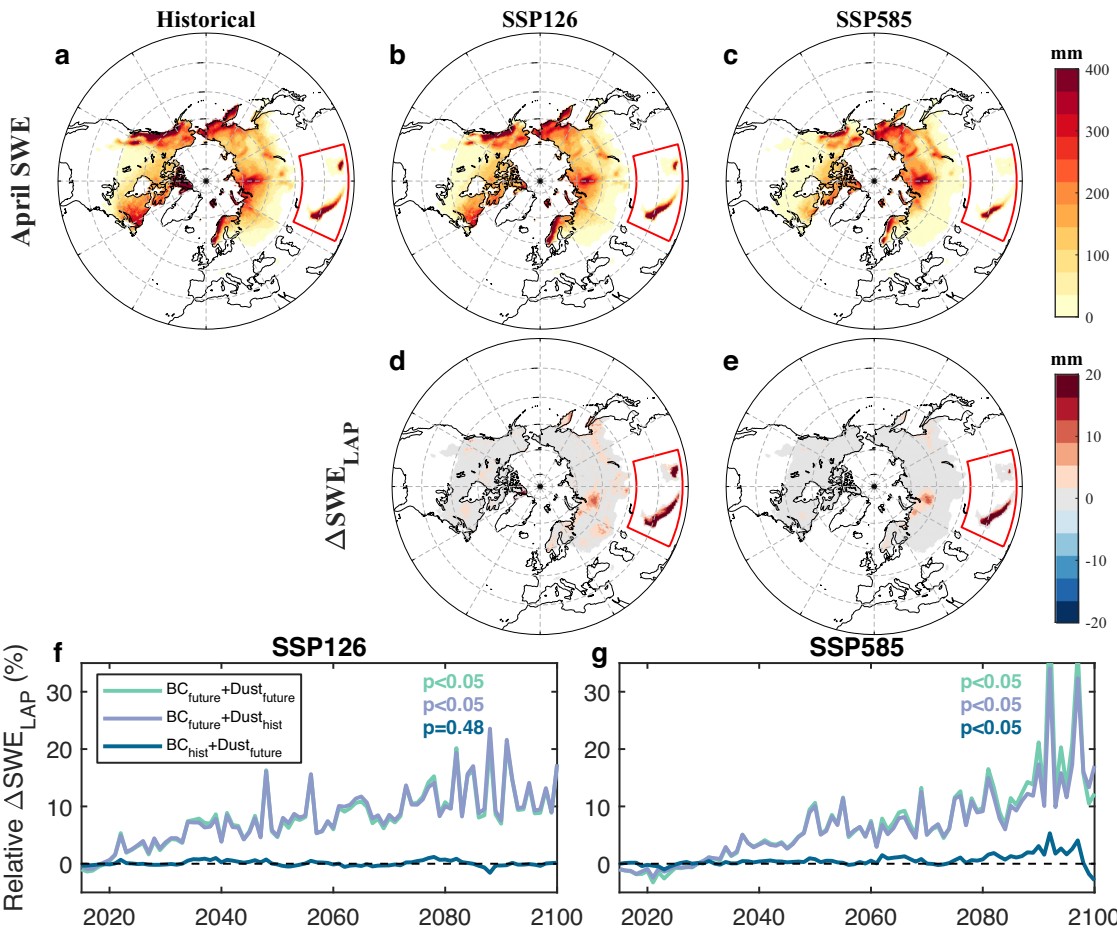

**Fig. 4 | Future snow water equivalent (SWE) in April and contributions of light-absorbing particles (LAP) change to future SWE. a** Historical (1995–2014) and **b, c** future (2081–2100) spatial patterns of SWE in April under SSP126 and SSP585. **d, e** The differences (ΔSWE_LAP) of future (2081–2100) April SWEs with and without LAP change under SSP126 and SSP585. **f, g** Time series of relative ΔSWE_LAP (calculated as the ratio of ΔSWE_LAP to projected SWE without LAP change) over snow-covered regions (where the average SWE exceeds 5 mm in the historical period) in the Tibetan Plateau (TP). In **a–e**, grids with an average April SWE smaller than 5 mm in the historical period are masked. In **f, g** BC_future + Dust_future, BC_future + Dust_hist, and BC_hist + Dust_future represent different combinations of black carbon (BC) and dust depositions, where the subscripts of future and hist represent future and historical average depositions, respectively. The *p* values from the Mann–Kendall (MK) test of statistical significance of the temporal trends from 2015 to 2100 are shown inside each panel.

still have large uncertainties when representing fire ignition, suppression, spread, and particularly emission, given our incomplete understanding of the complex and interacting controls on wildfire activities[42]. The uncertainties of wildfire simulations are expected to have only minor impacts on our results, as we focus on the snow season from December to May when wildfire activities are less frequent[43]. Although different ESMs show relatively similar magnitudes and spatio-temporal patterns of BC deposition (Fig. 1d, g), the uncertainties associated with the emission inventory, transport, and deposition modeling in ESMs should be quantified and reduced[14]. Furthermore, while ESMs reproduce the global spatial patterns and seasonal variations of dust distribution, there remain large differences in deposition rates (Fig. 1k, n) among ESMs due to uncertainties in simulating the dust life cycle (i.e., emission, transport, deposition). Such inconsistencies have been widely reported and are attributed to uncertainties of simulated land surface properties and atmospheric states[23,27]. Prior work shows that ESMs underestimate the amount of coarse dust with a diameter ≥5 μm[44]. Biological soil crusts (biocrusts), which are rarely considered in ESMs, have been found to have large effects on regional and global dust cycling[19]. These limitations could lead to an underestimation of dust impurity effects in both historical periods and future scenarios. However, our analyses are primarily based on relative differences rather than absolute values, mitigating

the impact of the previously mentioned uncertainties. The strong consistency between the simulations and observations of BC concentration in snow also confirms the reliability of our results (see the "Methods" section; Fig. S6). Advanced remote sensing observations, e.g., NASA's Earth Surface Mineral Dust Source Investigation (EMIT), are promising for providing potential constraints of the sign and magnitude for projecting dust deposition under climate scenarios.

Second, representing the evolutions of LAP and their mixing states with snow affects estimates of the LAP darkening effect on snow. The Snow, Ice, and Aerosol Radiative (SNICAR) model used in ELM (see the "Methods" section) can model the snow-darkening effects of BC and dust well and has been widely used in snow-related studies[45]. The Control ELM simulations assumed spherical snow grain shape, internal mixing of hydrophilic BC-snow, and external mixing of dust-snow (see the "Methods" section). Although non-spherical snow grain shape and the mixing state of LAP-snow can affect the magnitude of LAP-induced RF, their impacts on the relative contribution of future LAP change to RF is within ±2% (Fig. 3). More observations are needed to better constrain the irregular snow grain shape and space- and time-varying mixing states of LAP-snow in ESMs[34]. Although LAP scavenging processes via melting water can regulate LAP concentration after deposition, using either low or high melt-water scavenging efficiencies for hydrophilic BC leads to similar results in our simulations (Fig. 3).

Apart from BC and dust, brown carbon also has large darkening effects on snow[46]. However, it is not represented in most ESMs due to the large variations and high uncertainties of its chemical composition and optical properties. Snow algae also play important roles in snow melt[47], but nearly all ESMs neglect snow algae effects on snow. Although snow algae bloom and its distributions have been successfully implemented in the Minimal Advanced Treatments of Surface Interaction and Runoff Land Surface model (MATSIRO)[48], more observations and modeling are needed to evaluate and improve MATSIRO performance before further applications. Better constraining the projections of LAP impacts will benefit from the long-term field measurements and hyperspectral remote sensing satellite missions, e.g., the Surface Biology and Geology mission led by NASA.

Third, projecting future SWE change is sensitive to meteorological forcings and model configurations. Precipitation and air temperature largely determine the snowfall and snowmelt rates, producing large effects on snowpack[4]. However, air temperature and precipitation of ESM simulations still have systematic biases[49]. The Community Earth System Model Version 2 (CESM2) forcing data used in this study shows an overall consistent trend of future precipitation and air temperature projections over the TP with the ensemble mean of the analyzed CMIP6 models (Fig. S17a–d). Snowpack simulations are affected by the representations of various snow processes[50]. For example, topography affects the solar radiation received at the surface and snow processes[51], while land use and land cover change can also affect the snow accumulation and melting processes[52]. Despite their importance, both of these factors have small impacts on our analysis (Fig. 3). Although model uncertainties may influence SWE projections, our sensitivity experiments with different ELM model configurations show only small impacts on the relative contributions of LAP. Our previous study showed that ELM can well capture the snow distribution in the TP, compared to the MODIS remote sensing data, supporting the reliability of the results[34]. ELM simulations can reproduce the spatio-temporal pattern, interannual variability, and elevation gradient for different snow properties over the WUS[53]. The simulations are in line with the Snow Telemetry field measurements, MODIS remote sensing products, and data assimilation products[53]. The projected SWE trends of ELM simulations under both SSP126 and SSP585 are consistent with the ensemble mean of the seven CMIP6 models (Fig. S17e, f). These model sensitivity experiments, model evaluations, and intercomparisons provide confidence in simulating future SWE change using ELM.

With a focus on the future evolution of LAP deposition and RF over the NH, we identify a cleaner snow future and reduced snowpack loss, especially over the TP under both SSP126 and SSP585. We also expect cascading impacts of a cleaner snow future on environmental processes, socio-economic activities, and climate. For example, the future change of snowpack and snow phenology (i.e., evolution and duration) will potentially impact the mountain socio-ecological systems, e.g., the spring vegetation phenology[54] and hence, the terrestrial carbon cycle. The resulting reduction of snow loss in the future may alleviate the future threats to the downstream snowmelt-dependent agricultural production caused by global warming[2] but complicate future flood control and reservoir management. The reduced snow loss may slow down the future glacier retreat, considering that spring heavy snow could suppress summer glacier melt[55]. A cleaner snow future will also regulate the regional and global climate via snow-atmosphere coupling[56]. The air temperature will decrease with increased snow cover. Conceivably, the resulting reduction of snow loss over the TP will weaken surface heating, weaken vertical motion, strengthen the westerly jet stream, and eventually weaken the East Asian Summer Monsoon[57]. The snow cover change can also impact the magnitude, timing, and even sign of the South Asian Summer Monsoon and its precipitation[57]. The far-reaching implications of reduced LAP in snow for climate change and the underlying feedback mechanisms need further analysis via coupled ESM experiments.

## Methods

### CMIP6 simulations

Monthly aerosol deposition data during 1995–2100, including deposition rates of BC and dust from seven ESMs participating in CMIP6 (Table S2), are used in the study. We select the CMIP6 models that provide data for both BC and dust deposition with the variant of "r1i1p1f1". Two future scenarios of SSP126 and SSP585 are included in the analysis. All the data are remapped to a spatial resolution of 0.94° × 1.25° in latitude and longitude and are aggregated to the annual scale by calculating average values during December–May.

Only CESM2 provides aerosol deposition data with different BC and dust categories (i.e., hydrophilic BC, hydrophobic BC, dust particle size). The other six models just provided the total deposition rates of BC and dust. Therefore, we used the historical and future atmospheric $CO_2$ concentration, meteorological forcing data (e.g., precipitation, air temperature, humidity, wind speed, downward solar radiation, and downward longwave radiation), and aerosol deposition data from CESM2 to drive the ELM to further investigate future trends of LAP mass in snow.

### E3SM land model

The E3SM is a state-of-the-art fully-coupled ESM supported by the U.S. Department of Energy, which aims to improve and enhance actionable predictions of Earth system variability and change[58]. ELM, the land component of E3SM, originated from the Community Land Model Version 4.5 (CLM4.5)[59]. ELM can mechanistically simulate snow processes from snow accumulation to snow evolution to snow melt based on a multi-layer scheme. ELM can also prognostically simulate the change of LAP concentrations at different snow layers after deposition and how these changes affect snow albedo. Specifically, ELM uses a hybrid mode of SNICAR and the delta-Eddington adding-doubling radiative transfer solver to calculate the shortwave radiative characteristics of snow at different spectral bands[60]. The new SNICAR scheme in ELM treats both external mixing and internal mixing (within-hydrometeor) of BC and snow grains as well as size-dependent BC optical properties[61]. We recently extended ELM's ability to consider both external mixing and internal mixing between dust and snow grains as well as the impact of non-spherical snow grain shape on snow albedo[34]. In addition, we implemented a new parameterization that considers sub-grid topographic effects on solar radiation[51]. The model enhancements allow ELM to be used to investigate uncertainties related to model configurations on future snow projections. Our previous study showed that ELM can capture the spatio-temporal distribution and interannual variability of snow properties and timing compared to field measurements, remote sensing observations, and data assimilated products[34,53]. These confirm the effectiveness of simulating snow processes in ELM.

### Simulating the mass and RF of LAP in snow

To analyze future trends of the mass and RF of LAP, we conducted a series of offline ELM simulations at 0.5° spatial resolution over the NH from 1950 to 2100. The simulations were driven by CESM historical and future (SSP126 and SSP585) meteorological forcing and aerosol deposition data. We used the prescribed satellite phenology mode and downscaled the forcing data temporally and spatially to half-hourly and 0.5° resolution with bilinear interpolation methods. For each simulation, ELM was run at a half-hour time step with a monthly output frequency. The first 45 years (i.e., 1950–1994) were treated as model spin-up time and the remaining 106 years (i.e., 1995–2100) were used in the analysis.

To quantify the contribution of future LAP changes, we conducted a suite of simulations from 2015 to 2100 under the SSP126 and SSP585 scenarios with separate configurations of (1) monthly LAP deposition averaged in the historical period (1995–2014) (i.e., without interannual variation and trends in LAP); (2) future projected LAP deposition (i.e., with interannual variation and trends in LAP); (3)

historical average BC deposition and future projected dust deposition; and (4) future projected BC deposition and historical average dust deposition. To quantify the contribution of historical BC deposition, we also conducted an additional historical simulation without considering the snow-darkening effect of BC deposition. These simulations were carried out with the default ELM model settings, except that the simulations considered topographic effects on solar radiation.

To quantify uncertainties associated with model configurations, we also conducted ELM simulations with different model configurations related to snow grain shape, mixing state of LAP-snow, land use, and land cover change, topographic effects on solar radiation, and melt-water scavenging efficiency of hydrophilic BC (Table S3).

### Evaluation of ELM simulated BC concentration in snow

We gathered the historical field measurements of BC concentration in snow during 2000–2014 over the Arctic[62], TP[63–65], North China[66], and North America[67] as well as the snowpit measurements of both BC and dust concentration in snow over the TP[68,69] (Fig. S6a; Supplementary Data 1). The model-observation comparison (Text S1) shows that overall, the ELM simulations are statistically in good agreement with the observations for the BC concentration, especially in the snow column (Fig. S6b, c). The correlation coefficients are 0.36 and 0.66, respectively, for BC concentration in the top snow layer and the snow column. About 78% of the simulated BC concentration in the top snow layer is within a factor of four of the observed concentration, while that percentage is 83% for BC concentration in the snow column. The ELM simulations show an overestimation of BC concentration in the top snow layer, while such positive biases in ELM simulations are reduced for the BC concentration in the snow column, especially for the Arctic and North American sites. This may be, as suggested by ref. 67, because BC concentration in the snow column could be a better metric for model-observation comparisons as it smooths out the effects of new snowfall events and the variations in BC deposition rates, as well as the melting, percolation, and refreezing in-snow processes. Besides, the ELM simulated dust concentration in the snow column is well correlated to the snowpit measurements with a correlation coefficient of 0.74 (Fig. S6d). The simulated dust concentration for about 74% of the measurements is within a factor of four of the observed concentration. These results provide us confidence in ELM's ability to accurately estimating the LAP darkening effects on snow.

Note that the partial inconsistencies between observations and simulations are possibly caused by the spatial and temporal mismatch between the model simulations and field measurements[70]. The ELM simulations represent the average within a 0.5° coarse grid, while the field measurements are usually at a point scale. Thus, the spatial representativeness of the field measurements may be limited especially in summer with when the snow cover fraction is at its lowest and more heterogeneous. Considering that the sensitivity of snowpack to local topography is not well resolved by our simulations at 0.5° resolution and that the BC and dust concentrations in snow during spring may be representative of summer as well, we are doing our best to make use of the point-scale snowpit data to evaluate the ELM simulations, focusing more on the spatial variations than the absolute values for the specific periods and seasons. Besides, there are still also some uncertainties in field measurements related to the sample types, sampling snow depth, instrument errors, and measurement methods[14], which can significantly impact the model-observation comparison.

### Quantifying contributions of LAP changes to future snow-pack loss

Based on the ELM simulations, the difference ($\Delta SWE_{Climate}$) between the historical (1995–2014) SWE and projected future SWE (2081–2100) without LAP changes is used to characterize the impact of climate change (i.e., temperature and precipitation) (see Eq. (1)). The SWE

difference ($\Delta SWE_{LAP}$) between future SWE with and without LAP changes is used to represent the impact of future LAP changes (see Eq. (2)). We further define the ratio of $\Delta SWE_{LAP}$ to $\Delta SWE_{Climate}$ as the relative contribution of future LAP changes to the slowdown in snowpack loss ($\delta_{LAP}$), as shown in Eq. (3).

$$\Delta SWE_{Climate} = SWE_{Climate_{hist} + LAP_{hist}} - SWE_{Climate_{future} + LAP_{hist}} \quad (1)$$

$$\Delta SWE_{LAP} = SWE_{Climate_{future} + LAP_{future}} - SWE_{Climate_{future} + LAP_{hist}} \quad (2)$$

$$\delta_{LAP} = \frac{\Delta SWE_{LAP}}{\Delta SWE_{Climate}} \quad (3)$$

The mean and standard deviation of $\delta_{LAP}$ under different ELM configurations are used to represent the mean LAP effect and the corresponding uncertainty.

### Statistical analysis

All analyses are conducted using MATLAB R2019b (MathWorks Inc.). We use the non-parametric Mann–Kendall (MK) Tau test and Sen's slope to detect the monotonic trend of time series data. Trends with $p < 0.05$ are considered to be statistically significant in this study. Specifically, we use the 'ktaub' function from https://www.mathworks.com/matlabcentral/fileexchange/11190-mann-kendall-tau-b-with-sens-method-enhanced in this study.

## Data availability

Except for the CESM2 data, the aerosol emission, aerosol deposition, precipitation, and air temperature data in CMIP6 simulations can be freely downloaded from https://esgf-node.llnl.gov/search/cmip6/. All CESM2 data can be acquired using NCAR's data-sharing service. The ELM outputs from this study are openly available at https://doi.org/10.5281/zenodo.8253485 and https://github.com/daleihao/snow_SSP. The collected field measurements of BC and dust concentrations in snow are provided in the Supplementary Data 1 file.

## Code availability

The ELM code is publicly available at https://github.com/E3SM-Project/E3SM. Codes to generate all results and plot all figures are available at https://doi.org/10.5281/zenodo.8253485 and https://github.com/daleihao/snow_SSP.

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

## Acknowledgements

This research was supported by the U.S. Department of Energy (DOE), Office of Science, Biological and Environmental Research (BER) through the Earth System Model Development program area, as part of the Climate Process Team (CPT) and Energy Exascale Earth System Model (E3SM) projects. H.W. acknowledges support by the Regional and Global Model Analysis program area, as part of the HiLAT project. D.X. acknowledges support by the Earth System Modeling program area, as part of the ICoM project. This research was conducted at Pacific Northwest National Laboratory (PNNL), which is operated for the U.S. DOE by Battelle Memorial Institute under contract DE-AC05-76RL01830. This research used resources from the National Energy Research Scientific Computing Center (NERSC), a User Facility supported by the Office of Science of the U.S. DOE under contract no. DE-AC02-15 05CH11231, and computing resources from DOE BER Earth and Environmental Systems Modeling program's Compy cluster located at PNNL. We thank Shichang Kang and Yulan Zhan for sharing their BC-in-snow measurements over the TP. We thank Beth Mundy and Ben Bond-Lamberty for their editorial suggestions.

## Author contributions

D.H. and G.B. conceived the study. D.H. processed the data, conducted the simulations, performed the analysis, and drafted the original manuscript. D.H., G.B., H.W., D.X., H.H., Y.Q., and L.R.L. made suggestions to the design of the study and the analysis of the results and contributed to improving the manuscript.

## Competing interests

The authors declare no competing interests.
