## [Peer Review file · Nature Communications]

REVIEWER COMMENTS

Reviewer #1 (Remarks to the Author):

The manuscript entitled “A cleaner snow future mitigates Northern Hemisphere snowpack loss from warming” by Dalei Hao and co-authors investigated the effects of future change in LAP (BC & Dust) on aerosol-induced snow darkening, radiative forcing and change in SWE. Using a climate model coupled with a sophisticated land surface scheme, they have shown that cleaner snow will alleviate the loss of snowpack in the future due to a warmer climate. However, the results presented in this study are well-known. The BC emission over the developing countries will decrease in the future (or that is already in the decreasing phase) and hence the amount of BC aerosols deposited on the snow surface will decrease and we expect cleaner snow in the future. Compared to historic conditions, cleaner snow in the future will absorb less radiation and a positive change in SWE is expected. So the novelty of this study is not very clear.

If a reduction in BC could reduce 50% of the future snowpack loss over the Tibetan Plateau (Line no 18-20), what is the present-day contribution of BC to the observed snow cover loss? There are studies that reported comparable contributions of BC and dust on snowpack loss (Line no. 40). This gives the false impression that LAP is the sole contributor to the snow cover loss, irrespective of the warming due to greenhouse gases and LULC changes.

Most of the Himalayan glaciers are debris covered. How does BC deposition on snow influence glacier melt? Most of the references cited in this manuscript investigated the effects of aerosol deposition on snow cover, not on glaciers. Please modify the manuscript accordingly. Do you have clear evidence for the effect of aerosols on glaciers instead of seasonal snow?

Please provide the inter-comparison of simulated BC and observations over the distinct snow-covered regions, especially the Tibetan Plateau. Considering the uncertainties listed in line no. 247-270, what is the change in the contribution of BC-induced snow cover reduction? The flux of BC deposited on the snow surface is exorbitantly high. For example, to get $2 \text{ ng m}^{-2} \text{ s}^{-1}$ deposition flux at the surface, atmospheric BC concentration should be nearly $20 \text{ microgram m}^{-3}$. Such a high concentration of BC does not exist over the Tibetan region.

The BC RF at 2100 for SSP126 and SSP585 (Figure S4, solid lines: with future change of BC) are comparable. It shows that irrespective of the emission pathways (low or high), the BC forcing over the Tibetan Plateau remains the same in the future. Please explain this contradiction.

The authors did not explain the reason for the increase in SAE over the Karakoram region. Is this indicating a decrease in temperature or an increase in precipitation for SSP126? If so, why is this pattern missing in SSP585?

Most of the discussions in this manuscript were the seasonal mean of different variables from December to May. But for SWE, instead of considering the entire snow season, they have considered only April. Please explain. It looks like the change in SWE is not very significant for the entire snow season.

The authors highlighted in the abstract that “The reduced black carbon contamination in snow over the Tibetan Plateau will alleviate future snowpack loss due to climate change by $52.1 \pm 8.0\%$ and $8.0 \pm 1.1\%$ for the two scenarios.” The authors later stated in line no. 200-204 that this change is due to LAP (BC+Dust). Please clarify.

Minor comments:

Line no. 219: The cited reference does not report “the melting of snow and the retreat of glaciers due to LAP”.

Line no. 219-222: This is quite obvious. Anthropogenic emissions of LAP will decrease in the future, so the decrease in LAP deposited on the snow surface is of no surprise.

Figure S1: unit is not given.

Figure S2: Values reported in ng m^{-2} unit, but most of the in-situ measurements are in the unit of ng/g . Convert the unit used in all figures to units used for measurements.

Reviewer #2 (Remarks to the Author):

The authors estimate that the decreasing trend of the snow water equivalent caused by climate change will slow down in the future especially at the Tibetan Plateau. This is mainly due to a decrease in Black Carbon emissions. The Earth System Model ELM, is used for the calculation.

I find the authors have used appropriate techniques and made a thorough analysis and interpretation. They have presented the results in sufficient details and carefully discuss the method they have used.

I find the analysis is valid and the evidence for the conclusions are sufficiently strong. However, I find the study is rather narrow and would have liked to see a more comprehensive discussion on global climate aspects and maybe also a discussion on feedback mechanisms. The study has some significance, suggesting the findings could apply to other areas as well but also the presentation of the method will have some impact.

REVIEWER COMMENTS

Reviewer #1 (Remarks to the Author):

The manuscript entitled “A cleaner snow future mitigates Northern Hemisphere snowpack loss from warming” by Dalei Hao and co-authors investigated the effects of future change in LAP (BC & Dust) on aerosol-induced snow darkening, radiative forcing and change in SWE. Using a climate model coupled with a sophisticated land surface scheme, they have shown that cleaner snow will alleviate the loss of snowpack in the future due to a warmer climate. However, the results presented in this study are well-known. The BC emission over the developing countries will decrease in the future (or that is already in the decreasing phase) and hence the amount of BC aerosols deposited on the snow surface will decrease and we expect cleaner snow in the future. Compared to historic conditions, cleaner snow in the future will absorb less radiation and a positive change in SWE is expected. So the novelty of this study is not very clear.

Thanks for your constructive comments and suggestions. We have revised the manuscript carefully.

Sorry for the unclear statements on some research gaps filled by this study. We have reorganized and highlighted them as below:

1. **Quantify and understand the future evolution of LAP deposition and RF in snow.** Although the historical and present-day LAP impacts on snowpack have been widely recognized, how the LAP emission and deposition over snow-covered regions will change in the future and further impact snowpack in a warmer climate remains unknown. A clean or dirty snow future could have very different impacts on snowmelt and snow water resources with large societal implications.
2. **Quantify separately the relative contribution of climate change due to greenhouse gases and LAP evolution to future snowpack change.** Future global warming is projected to reduce the snow water resources globally. Simultaneously, the future change of LAP darkening effects due to changes in LAP emissions will also impact snow water availability. Understanding the relative contribution of climate change and LAP to future snowpack change under different SSP scenarios is critical for constraining projections of downstream freshwater availability from snowmelts.

3. Quantify the uncertainty in snowpack projection from the model configurations using the state-of-art E3SM Land Model (ELM).

Generally, climate models (e.g., CMIP6 models) predict a significant decrease of future global snowpack due to global warming. However, such snowpack projection can be biased because most climate models neglect or oversimplify the effects of snow darkening due to BC and dust deposition. The ELM can prognostically simulate the change of LAP concentrations in different snow layers after deposition and the impacts of these changes on snow albedo (Hao et al., 2022). By carrying out historical and future ELM simulations with different model configurations of topography, land use and land cover change, LAP-snow mix state, snow grain shape and BC meltwater scavenging efficiency, we quantified the uncertainties in snowpack projection associated with the model configurations.

We have clarified and strengthened these key points throughout the revised manuscript (especially in the abstract and introduction).

1. If a reduction in BC could reduce 50% of the future snowpack loss over the Tibetan Plateau (Line no 18-20), what is the present-day contribution of BC to the observed snow cover loss? There are studies that reported comparable contributions of BC and dust on snowpack loss (Line no. 40). This gives the false impression that LAP is the sole contributor to the snow cover loss, irrespective of the warming due to greenhouse gases and LULC changes.

Sorry for the confusion. Our results show that future LAP change in snow over the Tibetan Plateau will alleviate future snowpack loss due to climate change by $52.1 \pm 8.0\%$ and $8.0 \pm 1.1\%$ for the two scenarios, mainly due to reduced black carbon contamination. We stress that future BC deposition reduction will alleviate the snowpack change caused by global warming. To avoid the confusion, we have revised the abstract to make the statement clear.

We have also analyzed the historical ELM simulations to quantify the historical contribution of BC to snowpack by comparing the differences between the historical simulations with and without BC deposition (Figure R1). The results show that the deposited BC in snow reduces the historical Dec-May average SWE by up to 25-30 mm over the TP. Our findings are consistent with Ji (2016) that reported a decrease in average SWE in the non-monsoon season by up to more than 25 mm over the TP. Sarangi et al. (2020) revealed that the influence of dust

on snow darkening could be greater than that of BC over the high-latitude high-mountain Asia regions, and our results show a similar pattern (Figure R2). We have now added these results in Line 127-128 and 251-253 of the revised manuscript.

Figure R1| BC-induced reduction of historical (1995-2014) average SWE from December to May. The SWE reduction is calculated based on ELM historical simulations with and without BC deposition. The grids where the average SWE during December to May is smaller than 5 mm are masked.

Figure R2| Ratio of historical (1995-2015) average dust-induced albedo reduction to total albedo reduction over the TP. The snow albedo reduction is calculated based on ELM outputs from December to May. The grids where the average SWE during December to May is smaller than 5 mm are masked.

2. Most of the Himalayan glaciers are debris covered. How does BC deposition on snow influence glacier melt? Most of the references cited in this manuscript investigated the effects of aerosol deposition on snow cover, not on glaciers. Please

modify the manuscript accordingly. Do you have clear evidence for the effect of aerosols on glaciers instead of seasonal snow?

Thanks for the insightful comment. Considering that our study focuses on the LAP darkening effects on seasonal snowpack, we have deleted the irrelevant glacier-related citations and descriptions in the revised manuscript.

3. Please provide the inter-comparison of simulated BC and observations over the distinct snow-covered regions, especially the Tibetan Plateau. Considering the uncertainties listed in line no. 247-270, what is the change in the contribution of BC-induced snow cover reduction? The flux of BC deposited on the snow surface is exorbitantly high. For example, to get $2 \text{ ng m}^{-2} \text{ s}^{-1}$ deposition flux at the surface, atmospheric BC concentration should be nearly $20 \text{ microgram m}^{-3}$. Such a high concentration of BC does not exist over the Tibetan region.

Thank you for the good suggestion! We have added a comparison of simulated and observed BC concentration across the NH in Line 409-435 of the revised manuscript. Specifically, considering the temporal coverage of our historical simulations (1995-2014) and the availability of field observations, we have collected some field measurements of BC concentration in snow during 2000-2014 over the Arctic, Tibetan Plateau, North China, and North America from Doherty et al. (2010, 2014), He et al. (2018), Kang et al. (2022), Zhang et al. (2018) and Wang et al. (2013) (Figure R3a; Table S4 of the revised manuscript). The snow samples affected by drifting snow and with poor spatial representativeness reported in the relevant studies were excluded. For the estimates from the Integrating Sphere integrating SandWich (ISSW) spectrophotometer, we scaled the measured BC concentration in snow (Table S4 of the revised manuscript) to match the BC mass absorption efficiency of $7.5 \text{ m}^2 \text{ g}^{-1}$ used in ELM, following Qian et al. (2014). We collected information of both the BC concentration in the top snow layer and snow column. For the snow samples with unreported BC concentration in snow column, we averaged the surface and sub-surface BC concentration to get the approximate values. For the model evaluation, the snow samples within the same model grid cell for the same month and year were aggregated, and observations and simulations for the same geo-location, month, and year are compared.

Overall, the ELM simulations are statistically in good agreements with the observations for both the BC concentration in the top snow layer and snow column (Figure R3b,c). The correlation coefficients between the simulation and observation are 0.66 and 0.73, respectively, for BC concentration in the top snow layer and snow column. Overall, 78.4% of the simulated BC concentration in the

top snow layer is within a factor of four of the observed concentrations, and the number increases to 84.9% for BC concentration in the snow column. The ELM simulations show an overestimation of BC concentration in the top snow layer, while the positive biases in ELM simulations are smaller for the BC concentration in the snow column, especially for the Arctic and North America sites. As suggested by Doherty et al. (2014), BC concentration in snow column could be a better metric for model-observation comparisons because it smooths out the effects of new snowfall events, variations in BC deposition rates, and the melting, percolation, and refreezing processes after deposition in snow. Note that the partial inconsistencies between observations and simulations are also possibly caused by the spatial and temporal mismatch between simulations and field measurements, and uncertainties in field sampling and lab measurements of BC in meltwater. These results demonstrate that ELM can accurately estimate the LAP darkening effects on snow. We have added this evaluation in the Methods section of the revised manuscript.

Figure R3| Comparison of ELM-simulated and observed BC concentration in snow across the NH. a. Spatial distribution of field snow samples. **b,c** Scatter plots between observed and simulated BC concentration in the top snow layer and snow column. In (b,c), the dotted, dashed and solid lines are 1:1, 1:4 (or 4:1) and 1:10 (or 10:1) ratio lines, and the correlation coefficient and p value are labeled.

Regarding the impact of model uncertainty on the estimates of BC-induced snow cover reduction, the above-mentioned evaluation of the ELM simulations against the field observations confirms the reliability of ELM in simulating BC concentration. Our sensitivity analysis also demonstrated that the model configurations related to topography, land use and land cover change, LAP-snow mixing state, snow grain shape, and BC melt-water scavenging efficiency have small impacts on our conclusions. Besides, our analyses are primarily based on relative differences rather than absolute values, mitigating the impact of model uncertainties. We presented these in Line 273-317 of the revised manuscript.

Regarding the magnitude of BC deposition rate, we believe that the reviewer's estimation of atmospheric BC concentration of 20 microgram m^{-3} to get 2 $\text{ng m}^{-2} \text{s}^{-1}$

¹ BC deposition flux assumed only dry deposition of BC. However, both wet and dry depositions contribute to BC deposition. Dry deposition is mainly determined by the underlying surface characteristics and micrometeorological conditions and its velocity is generally within the range of $0.01\text{--}0.07\text{ cm s}^{-1}$ (Zhou et al., 2018). In contrast, BC wet deposition can be more efficient, depending on local precipitation (Figure R4). Wet deposition over the snow-covered regions account for over 80% of the total deposition (Figure R5). These ELM results are consistent with Barrett et al. (2019), He et al. (2014) and Textor et al. (2006). Besides, the field measurements at three stations in the Himalayas and southern TP show that the annual BC deposition rate can be $58.9\text{ mg}\cdot\text{m}^{-2}\cdot\text{year}^{-1}$, which is equal to $1.9\text{ ng m}^{-2}\text{ s}^{-1}$ (Table 2 in Yan et al., 2019). He et al. (2014) also showed that annual BC deposition can be larger than $10^{-5}\text{ kg m}^{-2}\text{ month}^{-1}$, which is equal to $3.8\text{ ng m}^{-2}\text{ s}^{-1}$. We have added Figure R4 in the revised manuscript.

Figure R4| Spatial patterns of historical precipitation over the Northern Hemisphere (NH). Here, the precipitation rate is calculated based on the ensemble mean of seven CMIP6 model outputs from December to May.

Figure R5| Spatial patterns of historical (1995-2014) BC deposition rate over the Northern Hemisphere (NH): a total, b wet, c dry deposition, and d the ratio of wet to total deposition. Here, the BC deposition rates are calculated based on the ensemble mean of seven CMIP6 model outputs from December to May. In each panel, grids with an average SWE from December to May smaller than 5 mm are masked.

4. The BC RF at 2100 for SSP126 and SSP585 (Figure S4, solid lines: with future change of BC) are comparable. It shows that irrespective of the emission pathways (low or high), the BC forcing over the Tibetan Plateau remains the same in the future. Please explain this contradiction.

Although SSP126 and SSP585 show different decreasing trends during 2015-2100, the BC RFs at the end of this century (2081-2100) are comparable between

SSP126 and SSP585 (0.57 and 0.61 W/m⁻², respectively). Despite the significant difference in the CO₂ emissions between SSP126 and SSP585, both scenarios present a significant BC emission reduction around the TP (Figure R6), which in turn leads to a significant reduction in the deposition of BC over the TP at the end of this century (Figure R7). This is because the main source of CO₂ and BC emission are different. For example, the energy sector tends to dominate the behavior of CO₂ emissions, while the residential commercial sectors (e.g., the biomass usage and mobile sources) overall dominate BC emissions across various future scenarios (Gidden et al., 2019; Xu et al., 2021).

The reduction of BC emission and deposition in the future is because of the reduction of BC emissions from the residential and commercial sector (e.g., biomass burning and motor vehicle diesel), which account for nearly 40 % of all BC emissions in the historical time period. However, at the end of the century, the contribution of the emissions from the residential and commercial sector is projected to decrease to a low level under both SSP126 and SSP585 due to a transition away from traditional biomass usage with the increased economic development and population stabilization and emissions controls on mobile sources (Gidden et al., 2019). For instance, BC emissions around the TP are the highest in China and India primarily due to traditional biomass usage in the residential sector and secondarily due to transport-related activity. In scenarios of high socioeconomic development and technological progress, such as SSP126 and SSP585, BC emissions across most countries decline dramatically by the end of the century. For instance, the total BC emissions in China are equal to those of the USA today. Note that even in SSP585, air pollutant emissions including sulfate and carbonaceous aerosols (i.e., organic and black carbon) are tightly controlled for environmental and health reasons (Kriegler et al., 2017). We added these explanations in Line 87-105 and 145-148 of the revised manuscript.

Figure R6| Spatial patterns of historical (1995-2100) and future (2081-2100) BC emission rates over the globe. Here, the BC emission rates are calculated based on the ensemble mean of seven CMIP6 model outputs from December to May.

Figure R7| Time series of average BC deposition from December to May over snow-covered Tibetan Plateau (TP) regions (where the average SWE exceeds 5 mm in the historical period of 1995-2014) under SSP126 and SSP585. The solid line and background shading represent the mean and standard deviation of BC deposition rates, respectively, based on the seven CMIP6 models.

5. The authors did not explain the reason for the increase in SAE over the Karakoram region. Is this indicating a decrease in temperature or an increase in precipitation for SSP126? If so, why is this pattern missing in SSP585?

Our results show that most TP regions will have increased precipitation and air temperature (Figure R8), as indicated by Yao et al. (2022). Different from other TP regions, the Karakoram region will have increased snowfall, which is consistent with Kapnick et al. (2014) and de Kok et al. (2018). This anomalous SWE phenomenon in the Karakoram could be linked to its unique seasonal cycle and winter precipitation, making the snow less sensitive to warming (Kapnick et al., 2014) and an increase in snowfall (de Kok et al., 2018; Yao et al., 2022). Consequently, the future SWE in Karakoram increases under SSP126 but stays relatively stable under SSP585. Compared to SS126, SSP585 projects significantly warmer air temperature (Figure R8h,i), leading to faster snowmelt than SSP126, which possibly explains the difference between these two scenarios. We have added these explanations in Line 203-207 of the revised manuscript.

Figure R8| Spatial patterns of historical and future precipitation, snowfall and air temperature over the TP. a,d,g Historical (1995-2014) spatial patterns of climate conditions. **b,c,e,f,h,i** The difference between future (2081-2100) and historical climate conditions under SSP126 and SSP585. Historical and future climate conditions are calculated based on CESM outputs from December to May.

6. Most of the discussions in this manuscript were the seasonal mean of different variables from December to May. But for SWE, instead of considering the entire snow season, they have considered only April. Please explain. It looks like the change in SWE is not very significant for the entire snow season.

Seasonal snowpack generally peaks around April in many NH mid-latitude snow-dominated regions, e.g., the Tibetan Plateau (Liu et al., 2021), Western US. The peak SWE is one of the metrics used to assess potential water supply outcomes (Kraaijenbrink et al., 2021). In this consideration, April SWE can provide useful insight into the expected spring runoff and inform reservoir operation and seasonal water supply forecasts that critically support agricultural and resource management decisions. Therefore, we selected April SWE in the analysis. We have added these explanations in Line 193-196 of the revised manuscript.

Additionally, we have computed the December-May average SWE, which shows the same patterns as those for April SWE (Figure R10 and R11), but with a relatively smaller magnitude due to the averaging effects. Specifically, for April SWE, the future LAP change in snow over the TP will alleviate future snowpack loss due to climate change by $52.1 \pm 8.0\%$ and $8.0 \pm 1.1\%$ for SSP126 and SSP585, respectively. For the December-May average SWE, those values are $33.3 \pm 4.5\%$ and $5.6 \pm 0.6\%$, respectively (Figure R10). We have added the corresponding figures on the December-May average SWE in the revised manuscript.

Figure R9| Spatial distribution of peak SWE timing (represented by day of water year (DOWY)). The inset figure is the histogram of peak SWE DOWY. The three dates labeled in the color bar (DOWY 133, DOWY 169 and DOWY 217) correspond to the 10th, 50th and 90th percentile in the DOWY distribution and are marked with vertical dashed lines in the inset histogram. This figure is cited from Liu et al. (2021).

Figure R10| Historical and future average SWE from December to May over the snow-covered TP regions (where the average SWE exceeds 5 mm in the historical period of 1995-2014) under different model configurations. For each panel, Climate_{hist}+LAP_{hist} represents the historical (1995-2014) simulations with historical LAP depositions, while Climate_{future}+LAP_{future} and Climate_{future}+LAP_{hist} represent future (2081-2100) simulations with and without a future change of LAP depositions, respectively. The Climate_{future}+LAP_{hist} simulations used the historical average LAP depositions from 1995-2014. The horizontal axis labels represent different model configurations (see **Methods**), where Control has the ELM default settings and the others represent major adjustments made from the Control case. Specifically, PP assumes that the terrain is flat and neglects topographic effects on solar radiation; Koch assumes a non-spherical snow grain shape (Koch snowflake); extBC assumes external mixing between hydrophilic BC and snow grains; intDust assumes internal mixing between dust and snow grains; noLULCC has no land use and land cover change; MSE_high assumes high melt-water scavenging efficiency

(MSE = 2, much higher than the default value of 0.2) of hydrophilic BC; and MSE_low assumes a low MSE (0.02) of hydrophilic BC. In (a,b), the contribution (δ_{LAP}) of future LAP change that mitigates snowpack loss under each ELM configuration is noted as a percentage and is calculated as the ratio of the SWE difference (ΔSWE_{LAP}) between Climate_{future}+LAP_{future} and Climate_{future}+LAP_{hist} to the SWE difference ($\Delta SWE_{Climate}$) between Climate_{hist}+LAP_{hist} and Climate_{future}+LAP_{future}.

Figure R11| Future average SWE from December to May and contributions of LAP change to future SWE. **a** Historical (1995-2014) and **b,c** future (2081-2100) spatial patterns of average SWE under SSP126 and SSP585. **d,e** The differences (ΔSWE_{LAP}) of future (2081-2100) SWEs with and without LAP change under SSP126 and SSP585. **f,g** Time series of relative ΔSWE_{LAP} (calculated as the ratio of ΔSWE_{LAP} to projected SWE without LAP change) over snow-covered regions (where the average SWE exceeds 5 mm in the historical period) in the TP. In (a-e), grids with an average SWE smaller than 5 mm in the historical period are masked. In (f,g), $BC_{future}+Dust_{future}$, $BC_{future}+Dust_{hist}$, and $BC_{hist}+Dust_{future}$ represent different combinations of BC and dust depositions, where the subscripts of future and hist represent future and historical average depositions, respectively.

7. The authors highlighted in the abstract that “The reduced black carbon contamination in snow over the Tibetan Plateau will alleviate future snowpack loss due to climate change by $52.1\pm 8.0\%$ and $8.0\pm 1.1\%$ for the two scenarios.” The authors later stated in line no. 200-204 that this change is due to LAP (BC+Dust). Please clarify.

Thank you for pointing out this inconsistency. Here the snowpack change is indeed caused by LAP (sum of BC and Dust) change. However, our results also show that the future reduction of BC concentration dominates the snowpack change in Figure 4f,g of the revised manuscript. Therefore, we have revised Line 18-21 in the abstract to “the projected LAP changes in snow over the Tibetan Plateau will alleviate future snowpack loss due to climate change by $52.1\pm 8.0\%$ and $8.0\pm 1.1\%$ at the end of the century for the two scenarios, mainly due to reduced black carbon contamination.”.

Minor comments:

1. Line no. 219: The cited reference does not report “the melting of snow and the retreat of glaciers due to LAP”.

Sorry for the wrong citation. As suggested in major comment #2 (Reviewer #1), we have deleted the less relevant, glacier-related citations in the revised manuscript, considering that our study focuses on the LAP darkening effects on seasonal snowpack.

2. Line no. 219-222: This is quite obvious. Anthropogenic emissions of LAP will decrease in the future, so the decrease in LAP deposited on the snow surface is of no surprise.

Indeed, the historical and present-day LAP impacts on snowpack have been widely recognized. However, the future evolution of LAP deposition and RF in snow are not well quantified. The relative contribution of climate change and LAP to future snowpack change remains unknown. We have clarified and strengthened these key points throughout the revised manuscript. Please also see our response to the overall comment from Reviewer #1 for details.

3. Figure S1: unit is not given.

We have added the units accordingly. This figure is for the relative trend of BC and dust deposition. We have also added a figure showing the absolute trend of BC and dust deposition in the revised manuscript.

4. Figure S2: Values reported in ng m^{-2} unit, but most of the in-situ measurements are in the unit of ng/g . Convert the unit used in all figures to units used for measurements.

Unfortunately, we did not output LAP concentration in the top snow layer with the units of (ng/g) in the ELM simulations. Thus, we have replaced it with the LAP concentration in snow column, which has the units of ng/g , in the revised manuscript. We have modified other figures accordingly in the revised manuscript.

References

- Barrett, T.E., Ponette-González, A.G., Rindy, J.E. and Weathers, K.C., 2019. Wet deposition of black carbon: A synthesis. *Atmospheric environment*, 213, pp.558-567.
- de Kok, R.J., Tuinenburg, O.A., Bonekamp, P.N. and Immerzeel, W.W., 2018. Irrigation as a potential driver for anomalous glacier behavior in High Mountain Asia. *Geophysical research letters*, 45(4), pp.2047-2054.
- Doherty, S.J., Warren, S.G., Grenfell, T.C., Clarke, A.D. and Brandt, R.E., 2010. Light-absorbing impurities in Arctic snow. *Atmospheric Chemistry and Physics*, 10(23), pp.11647-11680.
- Gidden, M.J., Riahi, K., Smith, S.J., Fujimori, S., Luderer, G., Kriegler, E., Van Vuuren, D.P., Van Den Berg, M., Feng, L., Klein, D. and Calvin, K., 2019. Global emissions pathways under different socioeconomic scenarios for use in CMIP6: a dataset of harmonized emissions trajectories through the end of the century. *Geoscientific model development*, 12(4), pp.1443-1475.
- Hao, D., Bisht, G., He, C., Bair, E., Huang, H., Dang, C., Rittger, K., Gu, Y., Wang, H., Qian, Y. and Leung, L.R., 2022. Improving snow albedo modeling in E3SM land model (version 2.0) and assessing its impacts on snow and surface fluxes over the Tibetan Plateau. *Geoscientific Model Development Discussions*, pp.1-31.
- He, C., Flanner, M.G., Chen, F., Barlage, M., Liou, K.N., Kang, S., Ming, J. and Qian, Y., 2018. Black carbon-induced snow albedo reduction over the Tibetan Plateau: uncertainties from snow grain shape and aerosol–snow mixing state based on an updated SNICAR model. *Atmospheric Chemistry and Physics*, 18(15), pp.11507-11527.
- He, C., Li, Q.B., Liou, K.N., Zhang, J., Qi, L., Mao, Y., Gao, M., Lu, Z., Streets, D.G., Zhang, Q. and Sarin, M.M., 2014. A global 3-D CTM evaluation of black carbon in the Tibetan Plateau. *Atmospheric Chemistry and Physics*, 14(13), pp.7091-7112.

- Ji, Z.M., 2016. Modeling black carbon and its potential radiative effects over the Tibetan Plateau. *Advances in Climate Change Research*, 7(3), pp.139-144.
- Kang, S., Zhang, Y., Chen, P., Guo, J., Zhang, Q., Cong, Z., Kaspari, S., Tripathee, L., Gao, T., Niu, H. and Zhong, X., 2022. Black carbon and organic carbon dataset over the Third Pole. *Earth System Science Data*, 14(2), pp.683-707.
- Kraaijenbrink, P.D., Stigter, E.E., Yao, T. and Immerzeel, W.W., 2021. Climate change decisive for Asia's snow meltwater supply. *Nature Climate Change*, 11(7), pp.591-597.
- Kriegler, E., Bauer, N., Popp, A., Humpenöder, F., Leimbach, M., Strefler, J., Baumstark, L., Bodirsky, B.L., Hilaire, J., Klein, D. and Mouratiadou, I., 2017. Fossil-fueled development (SSP5): An energy and resource intensive scenario for the 21st century. *Global environmental change*, 42, pp.297-315.
- Liu, Y., Fang, Y. and Margulis, S.A., 2021. Spatiotemporal distribution of seasonal snow water equivalent in High Mountain Asia from an 18-year Landsat–MODIS era snow reanalysis dataset. *The Cryosphere*, 15(11), pp.5261-5280.
- Qian, Y., Wang, H., Zhang, R., Flanner, M.G. and Rasch, P.J., 2014. A sensitivity study on modeling black carbon in snow and its radiative forcing over the Arctic and Northern China. *Environmental Research Letters*, 9(6), p.064001.
- Sarangi, C., Qian, Y., Rittger, K., Ruby Leung, L., Chand, D., Bormann, K.J. and Painter, T.H., 2020. Dust dominates high-altitude snow darkening and melt over high-mountain Asia. *Nature Climate Change*, 10(11), pp.1045-1051.
- Textor, C., Schulz, M., Guibert, S., Kinne, S., Balkanski, Y., Bauer, S., Berntsen, T., Berglen, T., Boucher, O., Chin, M. and Dentener, F., 2006. Analysis and quantification of the diversities of aerosol life cycles within AeroCom. *Atmospheric Chemistry and Physics*, 6(7), pp.1777-1813.
- Wang, X., Doherty, S.J. and Huang, J., 2013. Black carbon and other light-absorbing impurities in snow across Northern China. *Journal of Geophysical Research: Atmospheres*, 118(3), pp.1471-1492.
- Xu, H., Ren, Y.A., Zhang, W., Meng, W., Yun, X., Yu, X., Li, J., Zhang, Y., Shen, G., Ma, J. and Li, B., 2021. Updated global black carbon emissions from 1960 to 2017: improvements, trends, and drivers. *Environmental Science & Technology*, 55(12), pp.7869-7879.
- Yan, F., He, C., Kang, S., Chen, P., Hu, Z., Han, X., Gautam, S., Yan, C., Zheng, M., Sillanpää, M. and Raymond, P.A., 2019. Deposition of organic and black carbon: direct measurements at three remote stations in the Himalayas and Tibetan Plateau. *Journal of Geophysical Research: Atmospheres*, 124(16), pp.9702-9715.
- Yao T, Thompson L, Chen D, Chettri N., 2022. A Scientific Assessment of the Third Pole Environment.

- Zeng, X., Broxton, P. and Dawson, N., 2018. Snowpack change from 1982 to 2016 over conterminous United States. *Geophysical Research Letters*, 45(23), pp.12-940.
- Zhang, Y., Kang, S., Sprenger, M., Cong, Z., Gao, T., Li, C., Tao, S., Li, X., Zhong, X., Xu, M. and Meng, W., 2018. Black carbon and mineral dust in snow cover on the Tibetan Plateau. *The Cryosphere*, 12(2), pp.413-431.
- Zhou, J., Tie, X., Xu, B., Zhao, S., Wang, M., Li, G., Zhang, T., Zhao, Z., Liu, S., Yang, S. and Chang, L., 2018. Black carbon (BC) in a northern Tibetan mountain: effect of Kuwait fires on glaciers. *Atmospheric Chemistry and Physics*, 18(18), pp.13673-13685.

Reviewer #2 (Remarks to the Author):

The authors estimate that the decreasing trend of the snow water equivalent caused by climate change will slow down in the future especially at the Tibetan Plateau. This is mainly due to a decrease in Black Carbon emissions. The Earth System Model ELM, is used for the calculation. I find the authors have used appropriate techniques and made a thorough analysis and interpretation. They have presented the results in sufficient details and carefully discuss the method they have used. I find the analysis is valid and the evidence for the conclusions are sufficiently strong. However, I find the study is rather narrow and would have liked to see a more comprehensive discussion on global climate aspects and maybe also a discussion on feedback mechanisms. The study has some significance, suggesting the findings could apply to other areas as well but also the presentation of the method will have some impact.

We appreciate your comments and suggestions, and we have revised the manuscript accordingly!

Firstly, we have **reorganized the manuscript to strengthen and highlight the contributions of this study in the abstract and the introduction section**, including 1. **Quantify and understand the future evolution of LAP deposition and RF in snow**; 2. **Isolate the relative contribution of climate change and LAP evolution to future snowpack change**; and 3. **Quantify the uncertainty from the model configurations using the state-of-art E3SM Land Model (ELM)**. Please see our detailed response to Reviewer #1's comments for details.

Secondly, we have **added more discussion in Line 338-353 of the revised manuscript on the potential impacts of our findings on environmental processes, socio-economic activities, and climate to broaden the impacts of our study**. Specifically, we expect cascading impacts of a cleaner snow future on environmental processes, socio-economic activities, and climate. For example, the updated snowpack and snow phenology (i.e., evolution and duration) will potentially impact the mountain socio-ecological systems, e.g., the spring vegetation phenology (Wang et al., 2013) and thus terrestrial carbon cycle. The resulted increased availability of snow water resource may alleviate the future threats to the downstream snowmelt-dependent agricultural production caused by global warming (Biemans et al., 2019) and complicate future flood control and reservoir management. The increased snow cover may slow down the future glacier retreat (Painter et al., 2013). The cleaner snow future will also regulate the regional and global climate via snow-atmosphere coupling (Henderson et al.,

2018). The air temperature will decrease with the increased snow cover. Due to the complex atmospheric feedback, the increased or decreased snowfall can alleviate or aggravate future snowpack loss under a warming climate. Conceivably, the resulted increase of snow cover over the TP will weaken surface heating, and vertical motion, intensify the westerly jet stream and eventually weaken the East Asian Summer Monsoon (Li et al., 2018; You et al., 2020). The snow cover change can also impact the magnitude, timing, and even sign of the South Asian Summer Monsoon and its precipitation (You et al., 2020).

We stress that **the far-reaching implications of reduced LAP pollution in climate change and the corresponding feedback mechanism need further analysis via coupled ESM experiments**. We also urge more attentions on the future impacts of LAP on snow apart from climate change in Line 351-352 of the revised manuscript.

References

- Biemans, H., Siderius, C., Lutz, A.F., Nepal, S., Ahmad, B., Hassan, T., von Bloh, W., Wijngaard, R.R., Wester, P., Shrestha, A.B. and Immerzeel, W.W., 2019. Importance of snow and glacier meltwater for agriculture on the Indo-Gangetic Plain. *Nature Sustainability*, 2(7), pp.594-601.
- Doherty, S.J., Dang, C., Hegg, D.A., Zhang, R. and Warren, S.G., 2014. Black carbon and other light-absorbing particles in snow of central North America. *Journal of Geophysical Research: Atmospheres*, 119(22), pp.12-807.
- Henderson, G.R., Peings, Y., Furtado, J.C. and Kushner, P.J., 2018. Snow–atmosphere coupling in the Northern Hemisphere. *Nature Climate Change*, 8(11), pp.954-963.
- Li, W., Guo, W., Qiu, B., Xue, Y., Hsu, P.C. and Wei, J., 2018. Influence of Tibetan Plateau snow cover on East Asian atmospheric circulation at medium-range time scales. *Nature communications*, 9(1), p.4243.
- Painter, T.H., Flanner, M.G., Kaser, G., Marzeion, B., VanCuren, R.A. and Abdalati, W., 2013. End of the Little Ice Age in the Alps forced by industrial black carbon. *Proceedings of the national academy of sciences*, 110(38), pp.15216-15221.
- You, Q., Wu, T., Shen, L., Pepin, N., Zhang, L., Jiang, Z., Wu, Z., Kang, S. and AghaKouchak, A., 2020. Review of snow cover variation over the Tibetan Plateau and its influence on the broad climate system. *Earth-Science Reviews*, 201, p.103043.

Wang, T., Peng, S., Lin, X. and Chang, J., 2013. Declining snow cover may affect spring phenological trend on the Tibetan Plateau. *Proceedings of the National Academy of Sciences*, 110(31), pp.E2854-E2855.

REVIEWER COMMENTS

Reviewer #3 (Remarks to the Author):

This paper investigated the future spatio-temporal characteristics of LAP mass, snow albedo reduction, and surface radiative forcing induced by BC and dust, and separated the relative contribution of future climate change and LAP evolution to snowpack changes. The paper provides valuable information for future snowpack loss mitigation and policy-maker. The authors have done substantial revision and I am satisfied with reply letter.

The authors found that the projected LAP changes in snow cover over the Tibetan Plateau will alleviate future SWE loss due to climate change by $52.1 \pm 8.0\%$ and $8.0 \pm 1.1\%$ at the end of the century under SSP126 and SSP585, respectively, mainly due to reduced black carbon emission. Considering a large difference between two scenarios (green road and middle pathway), I think the significance of the work is for policy-makers that future green road pathway is great benefit for protection spring water supplies in Himalayan region and which is urgent to take an action. The authors should address this point in abstract and discussion section.

BC and dust dataset collected from snowpits (or snow column) in the upper glacier (not those from summer aged snow in glacier surface which experience strong melt) could be also useful for comparison with simulated BC and dust. There are two recent works provide such data in the Tibetan Plateau.

Yan F. et al., 2023. Dust dominates glacier darkening across majority of the Tibetan Plateau based on new measurements. *Science of the Total Environment*, 891, 164661.
<http://dx.doi.org/10.1016/j.scitotenv.2023.164661>.

Li Y., 2021. Black carbon and dust in the Third Pole glaciers: Revaluated concentrations, mass absorption cross-sections and contributions to glacier ablation. *Science of the Total Environment*, 789: 147746.
<https://doi.org/10.1016/j.scitotenv.2021.147746>.

Several minor comments are as follow.

1. In line 88-90 of the revised manuscript, "...using the state-of-the-art Energy Exascale Earth System Model (E3SM) Land Model (ELM) driven by meteorological and LAP changes simulated by a CMIP6 model (see Methods)". What are the changes in meteorological field? Please clarify it.
2. Line 100: Dust deposition increase under SSP126 and SSP585 in Himalaya and central Asia (Figure S2). Is this mainly due to anomaly atmospheric circulation or regional anthropogenic impacts in the future?

3. In line 101-104, the statement of "...emissions sources ranging from the significant fossil fuel combustion, traditional biomass usage, to transport-related activity in the Asian regions" seems not appropriate, since the transport emission is also a kind of source sector of fossil fuel combustion.
4. In line 203-205, the p values of the interannual variability of the average dust deposition are smaller than 0.05, which is inconsistent with those in and Fig. 1. Please check and revise it.
5. In line 257-258, "... due to the reduced LAP deposition (Fig. 1 and S4) and snowpack under climate change..." should be "...due to the reduced BC deposition..."
6. Line 346, I agree that the increased snow in spring may slow down the glacier retreat. One of reason is that spring heavier snow could protect summer glacier melt as indicated by Kang et al. (2009).
7. Table S1, please add unit.
8. The significance of trend of time series data was tested at the 95% confidence level ($p < 0.05$). The p value was presented in Fig.1 only and should be added in other figures also.
9. In the title of Fig.1, "... c,f,j,m their differences for (a-c,e-f) BC and (h-j,l-m) dust under SSP126 and SSP58" should be "...differences for (a-b, a-e) BC and (h-i, h-l) dust...". Please check details carefully.
10. "Consistent with Ref10, historically, dust plays a comparable role to BC in reducing snow albedo over high altitude regions of HMA (Fig. S7)" (line no. 244-245), but in Fig. S9, spatial patterns of historical snow albedo reduction caused by BC and dust are different obviously in this simulation.

REVIEWER COMMENTS

Reviewer #3 (Remarks to the Author):

This paper investigated the future spatio-temporal characteristics of LAP mass, snow albedo reduction, and surface radiative forcing induced by BC and dust, and separated the relative contribution of future climate change and LAP evolution to snowpack changes. The paper provides valuable information for future snowpack loss mitigation and policy-maker. The authors have done substantial revision and I am satisfied with reply letter.

Thanks for your constructive comments and suggestions. We have revised the manuscript carefully. Please see below for details.

The authors found that the projected LAP changes in snow cover over the Tibetan Plateau will alleviate future SWE loss due to climate change by $52.1 \pm 8.0\%$ and $8.0 \pm 1.1\%$ at the end of the century under SSP126 and SSP585, respectively, mainly due to reduced black carbon emission. Considering a large difference between two scenarios (green road and middle pathway), I think the significance of the work is for policy-makers that future green road pathway is great benefit for protection spring water supplies in Himalayan region and which is urgent to take an action. The authors should address this point in abstract and discussion section.

Thanks for the suggestions to improve the abstract and discussion section. Indeed, the contribution of future LAP changes can be very different under low and high emission scenarios. Cleaner snow can mitigate over half of the snowpack loss caused by climate change under the low emission scenario (i.e., SSP126). However, climate factors dominate snowpack loss under the worst emission scenario (i.e., SSP585). The change in the relative role of reduced LAP highlights the necessity of constraining global warming levels to mitigate snowpack loss. Compared to the high fossil-fuel development pathway (i.e., SSP585), a sustainable and green development pathway (i.e., SSP126) will alleviate future loss of water supply from snowmelt. We added more discussion on this in Line 258-265 of the revised manuscript.

Besides, we have also added the corresponding description in the abstract “Our findings highlight a cleaner snow future and its benefits for future water supply from snowmelt especially under the sustainable development pathway of SSP126” of the revised manuscript.

BC and dust dataset collected from snowpits (or snow column) in the upper glacier (not those from summer aged snow in glacier surface which experience strong melt) could be also useful for comparison with simulated BC and dust. There are two recent works provide such data in the Tibetan Plateau.

Yan F. et al., 2023. Dust dominates glacier darkening across majority of the Tibetan Plateau based on new measurements. *Science of the Total Environment*, 891, 164661.

<http://dx.doi.org/10.1016/j.scitotenv.2023.164661>.

Li Y., 2021. Black carbon and dust in the Third Pole glaciers: Reevaluated concentrations, mass absorption cross-sections and contributions to glacier ablation. *Science of the Total*

We appreciate your recommendation of the snowpit datasets for evaluating our simulations. We obtained the datasets from the two studies (illustrated in Figure R1a) and included them in Supplementary data 1. The model-observation comparison shows that overall, the ELM simulations are statistically in good agreements with the observations for the BC concentration especially in the snow column (Figure R1b,c). The correlation coefficients are 0.36 and 0.66, respectively, for BC concentration in the top snow layer and the snow column. About 78% of the simulated BC concentration in the top snow layer is within a factor of four of the observed concentrations, while that percentage is 83% for BC concentration in the snow column. Besides, the ELM simulated dust concentration in the snow column is well correlated to the snowpit measurements with a correlation coefficient of 0.74 (Figure R1d). About 74% of the simulated dust concentration in the snow column is within a factor of four of the observed concentrations. These results provide us with more confidence in using ELM to estimate the LAP darkening effects on snow. We have added these results in Line 416-433 and Supplementary Tet S1 of the revised manuscript.

Figure R1| Comparison of ELM-simulated and observed BC and dust concentration in snow across the NH. a. Spatial distribution of field snow samples. b,c Scatter plots between observed

and simulated BC concentration in the top snow layer and snow column. d. Scatter plots between observed and simulated dust concentration in the snow column. In (b,c,d), the dotted, dashed and solid lines are 1:1, 1:4 (or 4:1) and 1:10 (or 10:1) ratio lines, and the correlation coefficient and p value are labeled.

However, it should be noted that only the order-of-magnitude comparison is possible between climate model simulations and field measurements, considering that:

1) Spatial mismatch: Our model simulations are at relatively coarse resolution (0.5 degree), while the field measurements are usually at a small point scale. Considering the strong dependence of snowpack on the local topography and microclimate conditions, the spatial representativeness of field measurements may be limited especially for summer when snow cover fraction is at its lowest and more heterogeneous. Such issue has been widely recognized by the community (Qian et al., 2011, 2015; Kang et al., 2020). High spatial resolution simulations and multi-site observation networks are needed to resolve this scale mismatch issue.

2) Temporal mismatch: To utilize the snowpit data for model evaluation, we compared snowpit measurements over the TP available after 2014 (Li et al., 2021; Yan et al., 2023) with the climatological (2005-2014) average of the ELM historical simulations, since the ELM simulations after 2014 are driven by projected scenarios rather than the historically observed forcing. Furthermore, we compared the limited snowpit samples acquired during summer with the ELM simulated spring average values because the summer monthly outputs from ELM simulations show little snow cover during that period. Considering that the sensitivity of snowpack to local topography is not well resolved by our simulations at 0.5 degree resolution, and that the BC and dust in snow during spring may be representative of the same in summer, we are doing our best to make use of the point-scale snowpit data to evaluate the ELM simulations, focusing more on the spatial variations than the absolute values for the specific periods and seasons. As suggested by Qian et al. (2015), Yasunari et al. (2004) and Kang et al. (2020), more frequent snow impurity lifecycle measurements, e.g., at daily timescale are necessary for model validation in terms of different years and locations.

3) Snow depth mismatch: The snow depth and sampling thickness of the measurements may vary case by case. It is challenging to accurately match the vertical location of LAP concentrations in the model-observation comparison, primarily due to the lack of the detailed measurement information.

4) Uncertainties from field measurements: There are also some uncertainties in field measurements related to sample types, instrument errors, and measurement methods (Kang et al., 2020). Different studies measured snow samples with different surface snow types (e.g., fresh snow, aged snow, or granular ice) which can affect their inter-comparisons. BC concentrations in snow are primarily measured using laser-induced incandescence and thermal-optical methods. The former can determine the size distributions of BC particle, but its ability is limited by the instrument detection range (Lim et al., 2014). The accuracy of thermal-optical method can be affected by the presence of light-absorbing mineral dust in the snow samples (Li et al., 2017). As suggested by Kang et al. (2020), it is pressing to coordinate a LAP-in-snow measurement inter-comparison project to measure the same snow samples using different instruments/techniques,

with the goal of providing an optimal estimate (with quantified uncertainty) of the LAP-in-snow concentrations.

We have added more discussion on these uncertainties in model-observation intercomparison in Line 435-446 of the revised manuscript.

Kang S, Zhang Y, Qian Y, Wang H. A review of black carbon in snow and ice and its impact on the cryosphere. *Earth-Science Reviews* 210, 103346 (2020).

Li, C., et al. Re-evaluating black carbon in the Himalayas and the Tibetan Plateau: concentrations and deposition. *Atmos. Chem. Phys.* 17, 11899–11912 (2017).

Li Y, et al. Black carbon and dust in the Third Pole glaciers: Revaluated concentrations, mass absorption cross-sections and contributions to glacier ablation. *Science of the Total Environment* 789, 147746 (2021).

Lim, S., et al. Refractory black carbon mass concentrations in snow and ice: method evaluation and inter-comparison with elemental carbon measurement. *Atmospheric Measurement Techniques* 7 3307-3324 (2014).

Qian, Y., et al. Sensitivity studies on the impacts of Tibetan Plateau snowpack pollution on the Asian hydrological cycle and monsoon climate. *Atmos. Chem. Phys.* 11, 1929–1948 (2011).

Qian Y, et al. Light-absorbing particles in snow and ice: Measurement and modeling of climatic and hydrological impact. *Advances in Atmospheric Sciences* 32, 64-91 (2015).

Yan F, et al.. Dust dominates glacier darkening across majority of the Tibetan Plateau based on new measurements. *Science of the Total Environment* 891, 164661 (2023).

Yasunari, Teppei J., et al. The GOddard SnoW impurity module (GOSWIM) for the NASA GEOS-5 earth system model: Preliminary comparisons with observations in Sapporo, Japan. *Sola* 10, 50-56 (2014).

Several minor comments are as follow.

1. In line 88-90 of the revised manuscript, "...using the state-of-the-art Energy Exascale Earth System Model (E3SM) Land Model (ELM) driven by meteorological and LAP changes simulated by a CMIP6 model (see Methods)". What are the changes in meteorological field? Please clarify it.

Sorry for the misleading expression in the original manuscript. We used meteorological forcing and LAP deposition data to drive the ELM simulations. We modified it as "using the state-of-the-art Energy Exascale Earth System Model (E3SM) Land Model (ELM) driven by meteorological forcing and LAP deposition data simulated by a CMIP6 model" in Line 63-65 of the revised manuscript.

2. Line 100: Dust deposition increase under SSP126 and SSP585 in Himalaya and central Asia

(Figure S2). Is this mainly due to anomaly atmospheric circulation or regional anthropogenic impacts in the future?

Both climate change and human land use can contribute to the change of dust emission, transport, and deposition (Kok et al., 2023). For climate change, the future changes in soil moisture driven by precipitation, relative humidity, and surface wind have large impacts on the dust emission (Zhao et al., 2023). The change in atmospheric circulation and turbulent mixing can affect the long-range transportation of dust particles. The change in precipitation can also affect the wet deposition of dust. Future human land use change can contribute to the increase of dust emission by altering vegetation fraction (Tegen et al., 2004) and thus affect the dust deposition. We added the possible explanations in Line 101 and 267-269 of the revised manuscript. However, the complex attribution analysis on the dominant factors is beyond the scope of this study, which warrants further investigations.

Kok JF, et al. Mineral dust aerosol impacts on global climate and climate change. *Nature Reviews Earth & Environment* 4, 71-86 (2023).

Zhao Y, et al. Multi-model ensemble projection of global dust cycle by the end of 21st century using CMIP6 data. *Atmos Chem Phys Discuss* 2023, 1-28 (2023).

Tegen I, Werner M, Harrison SP, Kohfeld KE. Relative importance of climate and land use in determining present and future global soil dust emission. *Geophysical Research Letters* 31, (2004).

3. In line 101-104, the statement of "...emissions sources ranging from the significant fossil fuel combustion, traditional biomass usage, to transport-related activity in the Asian regions" seems not appropriate, since the transport emission is also a kind of source sector of fossil fuel combustion.

We agree. We have revised it as "emissions sources ranging from the significant fossil fuel combustion and traditional biomass usage in the Asian regions" in Line 78 of the revised manuscript.

4. In line 203-205, the p values of the interannual variability of the average dust deposition are smaller than 0.05, which is inconsistent with those in and Fig. 1. Please check and revise it.

Thanks for catching this inconsistency. Here the p values should be larger than 0.05. We have modified this error as "and have a small, and insignificant increasing trend (MK test: $p > 0.05$)" accordingly in 102-103 of the revised manuscript.

5. In line 257-258, "... due to the reduced LAP deposition (Fig. 1 and S4) and snowpack under climate change..." should be "...due to the reduced BC deposition..."

Thanks for pointing out this. We have modified it accordingly in Line 141-142 of the manuscript.

6. Line 346, I agree that the increased snow in spring may slow down the glacier retreat. One of reason is that spring heavier snow could protect summer glacier melt as indicated by Kang et al. (2009).

Good point! We have further explained the possible mechanisms in Line 350-352 of the revised manuscript: “The reduced snow loss may slow down the future glacier retreat, considering that spring heavy snow could suppress summer glacier melt⁵⁶”.

⁵⁶Kang S, et al. Early onset of rainy season suppresses glacier melt: a case study on Zhadang glacier, Tibetan Plateau. *Journal of Glaciology* 55, 755-758 (2009).

7. Table S1, please add unit.

We have added the radiative forcing (RF) unit of (W m^{-2}) accordingly in Table S1 of the revised manuscript.

8. The significance of trend of time series data was tested at the 95% confidence level ($p < 0.05$). The p value was presented in Fig.1 only and should be added in other figures also.

We have added the p values in all the figures related to the time-series analysis as well as the corresponding descriptions in the revised manuscript, except that in Figure S10, we introduced the p values in the figure caption because all the p-values of the temporal trends are smaller than 0.05.

9. In the title of Fig.1, “... c,f,j,m their differences for (a-c,e-f) BC and (h-j,l-m) dust under SSP126 and SSP58” should be “...differences for (a-b, a-e) BC and (h-i, h-l) dust...”. Please check details carefully.

To avoid the confusion, we have modified the figure caption as “their differences (calculated as Future - Historical) for BC and dust” in Line 111-112 of the revised manuscript.

10. “Consistent with Ref10, historically, dust plays a comparable role to BC in reducing snow albedo over high altitude regions of HMA (Fig. S7)” (line no. 244-245), but in Fig. S9, spatial patterns of historical snow albedo reduction caused by BC and dust are different obviously in this simulation.

Sorry for the misleading description. Indeed, our results show that the relative contributions of BC and dust to snow albedo reduction are regionally varied (Figs. S7 and S9). Sarangi et al. (2020) showed that the influence of dust on snow darkening is greater than that of BC at surface elevation above 4000 m over HMA. Our results also confirm the dominant role of dust over high-latitude regions of HMA. To make it clear, we have revised it as “Consistent with Ref¹⁰, historically, dust can play a greater role than BC in reducing snow albedo over high altitude regions of HMA (Fig. S7)” in Line 128-129 of the revised manuscript.

Sarangi C, et al. Dust dominates high-altitude snow darkening and melt over high-mountain Asia. *Nature Climate Change* 10, 1045-1051 (2020).